# Molecular basis of VEGFR1 autoinhibition at the plasma membrane

Manas Pratim Chakraborty[1], Diptatanu Das [1], Purav Mondal [1], Pragya Kaul [1], Soumi Bhattacharyya[1], Prosad Kumar Das [1] & Rahul Das [1,2] ✉

Ligand-independent activation of VEGFRs is a hallmark of diabetes and several cancers. Like EGFR, VEGFR2 is activated spontaneously at high receptor concentrations. VEGFR1, on the other hand, remains constitutively inactive in the unligated state, making it an exception among VEGFRs. Ligand stimulation transiently phosphorylates VEGFR1 and induces weak kinase activation in endothelial cells. Recent studies, however, suggest that VEGFR1 signaling is indispensable in regulating various physiological or pathological events. The reason why VEGFR1 is regulated differently from other VEGFRs remains unknown. Here, we elucidate a mechanism of juxtamembrane inhibition that shifts the equilibrium of VEGFR1 towards the inactive state, rendering it an inefficient kinase. The juxtamembrane inhibition of VEGFR1 suppresses its basal phosphorylation even at high receptor concentrations and transiently stabilizes tyrosine phosphorylation after ligand stimulation. We conclude that a subtle imbalance in phosphatase activation or removing juxtamembrane inhibition is sufficient to induce ligand-independent activation of VEGFR1 and sustain tyrosine phosphorylation.

The vascular endothelial growth factor receptors (VEGFR) are the key regulator of normal physiological and pathological angiogenesis and vasculogenesis[1,2]. The VEGFR family comprises three receptor tyrosine kinases (RTK): VEGFR1, VEGFR2, and VEGFR3. Among them, VEGFR1 is an elusive family member. Even after three decades of its discovery, the function and regulation of VEGFR1 remain poorly understood[3–5]. VEGFR2 is the primary receptor for VEGFs that regulates diverse cellular functions, including blood vessel development during embryogenesis, hematopoiesis, and tumor angiogenesis[1,6]. VEGFR3, on the other hand, is the primary receptor for the lymphangiogenic factor VEGF-C and VEGF-D[2]. During embryonic development, VEGFR1 acts as a decoy receptor. The VEGFR1 and VEGFR2 are activated by a common bivalent ligand (VEGF-A). VEGFR1 negatively regulates the VEGFR2 signaling by sequestering excess VEGF-A, preventing overactivation of VEGFR2[5,7,8] Compared to VEGFR2, VEGFR1 binds to its ligand VEGF-A with a ten-fold stronger affinity[9,10]. Yet, the ligand binding induces only a weak kinase activation in VEGFR1 and does not

generate subsequent downstream signaling in endothelial cells, vascular smooth muscle, or fibroblast cells[4,8,11,12]. Although VEGFR1 and VEGFR2 share a high degree of sequence and structural homology, it is unclear why the two RTK are differently regulated.

VEGFR1 and VEGFR2 share similar structural architecture, comprising an extracellular ligand-binding domain (ECD) made up of seven immunoglobulin-like subdomains (D1 to D7), a single-passed transmembrane (TM) segment, a cytosolic juxtamembrane (JM) segment tethered to a kinase domain (KD) followed by a C-terminal tail (Fig. 1a)[1,2,13]. VEGFRs are activated by a canonical model of monomer to dimer (or multimer) transition upon ligand binding[14–16]. In the unligated state, the receptor exists predominantly as a monomer (Fig. 1b)[17,18], and the kinase-domain adopts a platelet-derived growth factor receptor (PDGFR)-like JM-in inactive conformation[19–22]. In the inactive conformation, the N-terminal portion of the JM segment is buried inside the catalytic site, like a wedge, locking the kinase domain in an autoinhibited conformation. Binding to bivalent ligands leads to a

[1]Department of Biological Sciences, Indian Institute of Science Education and Research Kolkata, Mohanpur campus, Mohanpur 741246, India. [2]Centre for Advanced Functional Materials, Indian Institute of Science Education and Research Kolkata, Mohanpur campus, Mohanpur 741246, India. ✉e-mail: rahul.das@iiserkol.ac.in

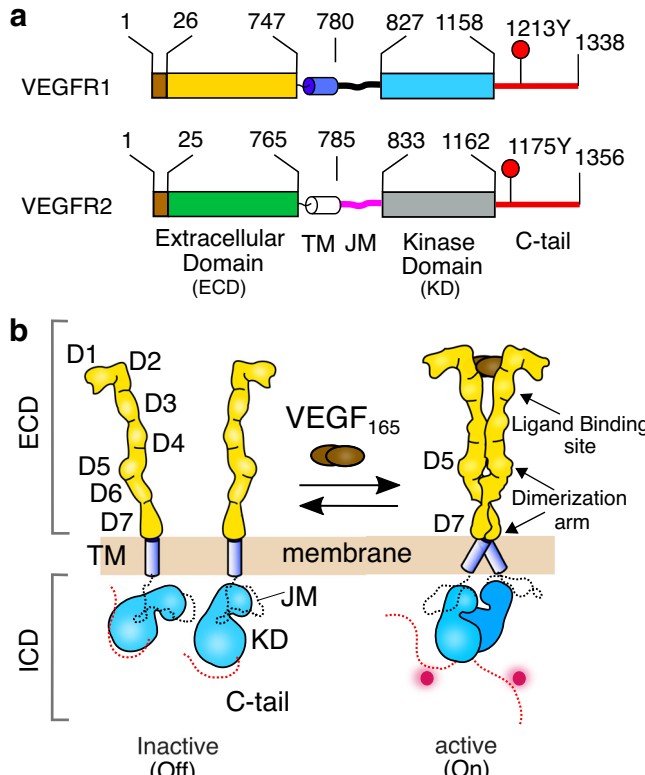

**Fig. 1 | Activation model of VEGFR. a** Schematic representation of domain architecture of VEGFR1 and VEGFR2. The transmembrane and juxtamembrane segments are labelled TM and JM, respectively. The C-terminal phosphotyrosine residues used for probing kinase activation are labelled. **b** Classical model of VEGFR activation in the presence of ligand (VEGF$_{165}$)[26]. The schematics are made using Inkscape Ver 1.2 See Supplementary Fig. 1.

ligand-dependent dimerization of the ECD (Fig. 1b)[23,24]. In the ligated state, the VEGFR forms a symmetric dimer in the outside (ECD) and an asymmetric dimer inside (kinase domain)[23–25]. The ECD dimerization rearranges the TM segment[26], removes the JM-inhibition (to JM-out conformation), and brings two adjacent kinase-domain in close proximity, allowing autophosphorylation of multiple tyrosine residues in the C-terminal tail (Fig. 1b)[19,21,23,27]. The phosphotyrosine residues then function as a docking site for assembling downstream signaling modules. Structural analysis of the kinase domain suggests that VEGFR1 is not a pseudokinase. All the regulatory motifs (R-spine and C-spine) and the catalytic residues are conserved (Supplementary Fig. 1a)[28,29]. The lack of kinase activity of VEGFR1 was attributed to an inhibitory sequence in the JM segment[30] and Asn1050 in the A-loop[31], the molecular mechanism of which is unknown.

In contrast, VEGFR1-signaling is indispensable in regulating hematopoietic cell function and developing pathophysiological conditions[5,7]. Ligand-dependent activation and VEGFR1-mediated cell-signaling regulate diverse physiological functions[32–35]. Overexpression or downregulation of VEGFR1 is linked to several cancers and cancer-associated pain, retinopathy, tumor survival, and autoimmune disorders[36–42]. The mechanism of how VEGFR1 autoinhibition is released under pathological conditions is an open question.

To gain further insight, we investigated the ligand-independent and ligand-dependent activation of VEGFR1 and VEGFR2 on the plasma membrane by a single-cell assay using fluorescence microscopy. Our data revealed that, unlike VEGFR2, VEGFR1 does not show concentration-dependent autophosphorylation in the absence of a ligand and is transiently phosphorylated upon ligand stimulation. We decipher that an electrostatic latch in the JM-S and an H-bond between

a tyrosine residue in the JM-B and C-helix in VEGFR1 together constitute a JM inhibition that likely stabilizes the inactive JM-in conformation. Slow release of the JM inhibition makes the VEGFR1 autophosphorylation inefficient. Finally, we proposed a mechanism explaining how a delicate balance in releasing JM inhibition maintains the VEGFR1 signaling constitutively off in the unligated state.

## Results

### VEGFR1 does not show concentration-dependent activation without ligands at the plasma membrane

Ligand-independent activation of RTKs is a key signature of several forms of cancer and manifestation of drug resistance[43–49]. Receptor density at the plasma membrane is an important determinant of ligand-independent activation of RTKs[50–57]. The density-dependent activation of RTK was explained by an equilibrium shift model between multiple receptor species[58,59]. Recent studies showed that VEGFR2 forms a ligand-independent dimer at a physiological concentration on the membrane and is able to autophosphorylate[26]. We ask, in the unligated state, if VEGFR1 autophosphorylates spontaneously on the plasma membrane.

We begin with a single-cell assay to comparatively study the concentration-dependent activation of VEGFR1 and VEGFR2 at the plasma membrane with and without ligand stimulation, respectively (Fig. 2, Supplementary Figs. 1, 2). We transiently transfected CHO cell lines with VEGFR1-mCherry or VEGFR2-mCherry constructs and stimulated them with VEGF$_{165}$. The transient transfection generates a heterogeneous population of cells expressing a diverse concentration of receptors on the plasma membrane. Since the localization of VEGFR family kinases is not solely restricted to the plasma membrane[60,61], in this study, we focused on the regions surrounding the plasma membrane (cell perimeter) (Supplementary Fig. 1d). The activation of the receptor at the membrane was probed by determining the phosphorylation level of Y1213 or Y1175 for VEGFR1 or VEGFR2, respectively, with specific antibodies[62–64]. We observed that the unligated VEGFR2 did not autophosphorylate Y1175 at low receptor concentrations but phosphorylates spontaneously at higher receptor concentrations[59] (Fig. 2a, b, e). Our data suggests that a critical concentration of the receptors at the plasma membrane is required to activate the kinase domain ligand independently. Whereas, upon ligand stimulation, VEGFR2 linearly phosphorylates Y1175, suggesting the phosphorylation is independent of receptor concentration (i.e., phosphorylation level is proportional to the receptors at the plasma membrane).

In the single-cell assay, the VEGFR1 also linearly phosphorylates Y1213 when stimulated with VEGF$_{165}$ (Fig. 2d, f). However, the fraction of tyrosine phosphorylated by VEGFR1 is significantly lower than VEGFR2 (Supplementary Fig. 2f). Unexpectedly, we observed that VEGFR1 did not show any ligand-independent autophosphorylation of Y1213, even at the highest receptor concentration measured in our studies (Fig. 2c, f). To rule out if the lack of ligand-independent activation of VEGFR1 is not cell-dependent, we repeat the assay by transiently transfecting VEGFR1 to COS-7 and a macrophage cell line (RAW264.7) (Supplementary Fig. 2a–e). We observed a similar phosphorylation profile of Y1213, as seen in the CHO cell line (Fig. 1f and Supplementary Fig. 2c–d). Suggesting that the lack of ligand-independent activation of VEGFR1 is an intrinsic property of the receptor and not an artifact. We then ask: why does VEGFR1 phosphorylate a lower fraction of tyrosine residues than VEGFR2 upon ligand stimulation? What is the molecular basis that constitutively inactivates VEGFR1 in the ligand-free state?

### Phosphorylation of VEGFR1 is transiently stable compared to VEGFR2

To understand why VEGFR1 and VEGFR2 are phosphorylated differentially upon ligand stimulation, we next studied the phosphorylation kinetics and half-life of phosphotyrosine residue (Fig. 2g, h and

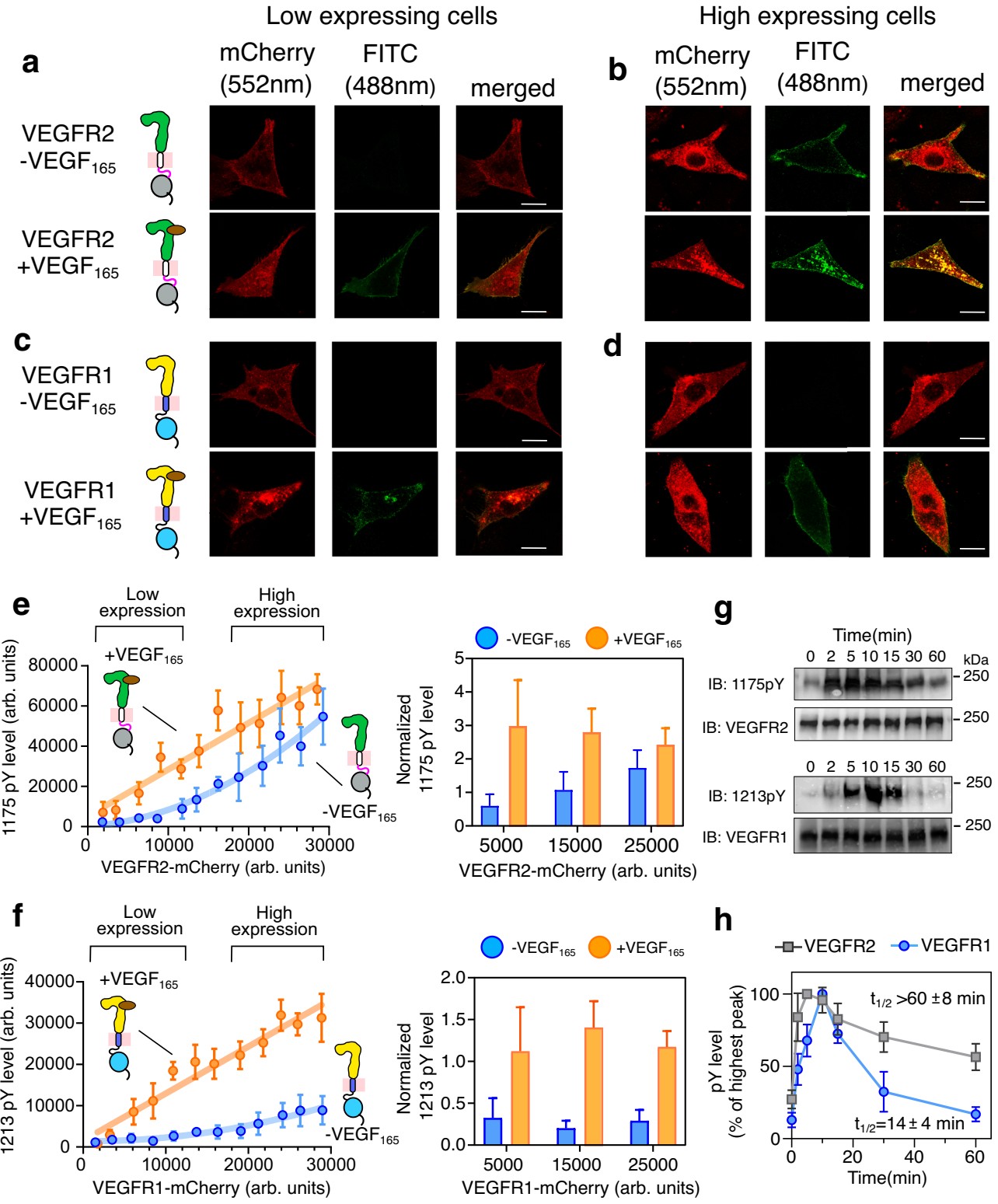

Supplementary Fig. 2g, h). We determined the phosphorylation level of Y1213 or Y1175 in VEGFR1 or VEGFR2, respectively, by immunoblotting over a period of time after ligand stimulation. The phosphorylation kinetics (Fig. 2h, Supplementary Fig. 2g, and Supplementary Table 4) shows that VEGFR1 is phosphorylated slower (rate = $0.07 \pm 0.01$ arb. units/min) than VEGFR2 (rate = $0.17 \pm 0.02$ arb. units /min). We also note that the phosphotyrosine (Y1213) in VEGFR1 is transiently stable ($t_{1/2} = 14 \pm 4$ min) compared to sustained phosphorylation of Y1175 in VEGFR2 ($t_{1/2} > 60 \pm 8$ min) (Fig. 2h, Supplementary Fig. 2i and

Supplementary Table 4). In VEGFR1, Y1213 phosphorylation peaks approximately at ten minutes, and the phosphorylation decays by forty minutes. In contrast, in VEGFR2, the phosphorylation of Y1175 peaks approximately at five minutes, and the phosphorylation does not decay to fifty percent by one hour. We next determine the phosphorylation kinetics of the total phosphotyrosines to find out if the phosphorylation pattern of VEGFR1 and VEGFR2 is not specific to the Y1213 and Y1175 but a general property of the respective VEGFR isotype (Supplementary Fig. 1b). We observed that the decay kinetics of the

**Fig. 2 | Measurement of ligand-independent and dependent VEGFR activation on the plasma membrane. a–d** Confocal images of VEGFR2 or VEGFR1 fused to mCherry in a low (**a, c**) and high (**b, d**) expressing CHO cell lines. The VEGFR expression level is shown in red ($\lambda_{ex}$ = 552 nm, $\lambda_{em}$ = 586-651 nm), and the phosphorylation status is shown in green ($\lambda_{ex}$ = 488, $\lambda_{em}$ = 505-531). Scale bar = 10 μm. **e, f** The expression level of VEGFR2 (panel e) or VEGFR1 (panel f) is plotted against the phosphorylation level of the corresponding tyrosine residues at the C-terminal tail. The low-expressing and high-expressing cells are indicated based on the mCherry intensity at the plasma membrane. Individual data points in the left panel represent the mean expression and phosphorylation level for the binned cells. The orange line represents the linear fitting of the individual data points in the ligand-dependent activation. The blue line in panel e represents the second-order polynomial fitting of the individual data points in the ligand-independent activation. In panel f, the blue line is the guiding line. The right panel represents the bar plot of the normalized phosphotyrosine levels. The phosphotyrosine level (FITC channel) is normalized with respect to the corresponding VEGFR expression level (mCherry

channel) at the plasma membrane. In (**e**) (left), $n$ = 85 (VEGFR2-VEGF$_{165}$), 89 (VEGFR2 + VEGF$_{165}$), and in (**f**) (left) $n$ = 107 (VEGFR1-VEGF$_{165}$), 100 (VEGFR1 + VEGF$_{165}$) cells were examined over five independent experiments in (**e, f**) (right) Each bar represents the mean value of 30–40 cells in the bar plot. The error bar shows the standard deviation of data points. Data are presented as mean values ± SD from five independent experiments. **g** The immunoblot shows the representative phosphorylation level of VEGFR1 or VEGFR2 at the indicated time points after activating the transfected CHO cell line with 50 nM VEGF$_{165}$. ($n$ = 3). **h** The plot of the phosphorylation level of respective C-terminal tyrosine residue as a function of time. The phosphorylation level is analyzed from the densitometric measurement of the Western blot shown in (**g**). The $t_{1/2}$ is determined by fitting the decay of the highest intensity observed to exponential decay. Data are presented as mean values ± SD from three independent experiments. All data were plotted using GraphPad Prism Ver 9.5.1. The confocal images were generated using Fiji Ver 1.54 f. The schematics were made using Inkscape Ver 1.2. Source data are provided as a Source Data file for panels e-h. See Supplementary Figs. 1 and 2.

total phosphotyrosines follow the same pattern as of Y1213 and Y1175 in VEGFR1 and VEGFR2, respectively. We speculate that the slow phosphorylation rate and transient stability of phosphotyrosine residue in VEGFR1 may contribute to a lower fraction of phosphorylated tyrosine residue (Supplementary Fig. 2f). We ask why the VEGFR1 phosphorylation is transient.

### Deletion of ECD does not constitutively activate the VEGFR1

The ligand-independent activation of VEGFR is obstructed by electrostatic repulsion between the Ig-like domain (D4-7) in the ECD dimer interface (Supplementary Fig. 3a)[65–67]. Despite that, substituting cysteine at 482 with arginine (C482R mutation) in the D5 of VEGFR2, linked to infantile hemangioma[68], constitutively activates the kinase by stabilizing a ligand-independent dimer[26]. A similar pathogenic cysteine to arginine substitution was reported for fibroblast growth factor receptors (FGFR)[54,69]. This suggests that a conserved ligand-independent activation mechanism prevails in RTKs carrying similar Ig-like ECD fold. However, no such mutation has been reported for VEGFR1. We next investigated if mutating the homologous C482 to arginine constitutively activates VEGFR1. We replaced C471R in the D5 of VEGFR1 and determined its activation (Fig. 3a, b). As expected, the VEGFR2 C482R mutant is constitutively activated and linearly phosphorylates Y1175 even without a ligand (Fig. 3c and Supplementary Fig. 3c, e). Surprisingly, the VEGFR1 C471R mutant, in the unligated state, is constitutively autoinhibited (Fig. 3d and Supplementary Fig. 3b, d). We wonder if the inability to dimerize renders the VEGFR1 C471R mutant inactive.

We next study the oligomeric states of VEGFR1 and VEGFR2 by fluorescence recovery after photobleaching (FRAP) experiment (Fig. 3e and Supplementary Fig. 4)[70]. The oligomeric status of VEGFR constructs was determined from the diffusion coefficient ($D_{confocal}$) derived from the rate of fluorescence recovery at the bleached spot on the plasma membrane. We considered that the dimeric state would have slower $D_{confocal}$ relative to the monomeric state[55]. In our experiment, we used two chimeric constructs of VEGFR1 as a monomer (named VEGFR1-GPA-G83I) and dimer (named VEGFR1-GPA) control, where the TM helix is replaced by glycophorin-A (GPA) G83I mutant and wild-type GPA, respectively (Fig. 3a)[71,72]. The relative increase in the $D_{confocal}$ for the VEGFR1-GPA-G83I mutant (0.033 ± 0.013 μm²s⁻¹) confirms a dimer-to-monomer transition on mutating TM segment in the VEGFR1-GPA chimera (0.021 ± 0.005 μm²s⁻¹) (Fig. 3e, Supplementary Fig. 4g, and Supplementary Table 3). We first turned to VEGFR2 to determine the $D_{confocal}$ for the wild-type and C482R mutant in the presence and absence of VEGF$_{165}$, respectively (Fig. 3e, Supplementary Fig. 4e, f, and Supplementary Table 3). Overall, $D_{confocal}$ for VEGFR2 agrees with the recently published data[26]. The wild-type VEGFR2 ($D_{confocal}$ = 0.021 ± 0.008 μm² s⁻¹) tends to form a ligand-independent

dimer, which explains the concentration-dependent activation of VEGFR2 in the absence of ligand (Fig. 2e). The ligand binding reorients the ECD and induces oligomerization ($D_{confocal}$ = 0.011 ± 0.004 μm² s⁻¹) mediated by a homotypic interaction between D4, D5, and D7 (Fig. 3e, Supplementary Figs. 3a, and 4c–f)[17,23,24]. Our data shows that the VEGFR2 C482R mutant forms a stable ligand-independent dimer ($D_{confocal}$ = 0.017 ± 0.006 μm² s⁻¹) that spontaneously activates the kinase domain (Fig. 3c, e and Supplementary Table 3)[26].

We then evaluated the dimerization propensity for the VEGFR1 constructs and made the following observations (Fig. 3e, Supplementary Figs. 4, 5a–b, and Supplementary Table 3). 1) The VEGFR1 does not dimerize in the absence of a ligand ($D_{confocal}$ = 0.038 ± 0.018 μm² s⁻¹), and the ligand binding induces receptor dimerization ($D_{confocal}$ = 0.018 ± 0.007 μm² s⁻¹) (Fig. 3e, and Supplementary Fig. 4a, b, e, f). 2) Formation of the ligand-dependent dimer is independent of kinase activity as suggested by the $D_{confocal}$ (0.016 ± 0.006 μm² s⁻¹) of the kinase-dead mutant (D1022N) (Supplementary Fig. 5a-b). 3) The C471R mutation in the ECD of the VEGFR1 does not induce spontaneous dimerization ($D_{confocal}$ = 0.035 ± 0.014 μm² s⁻¹). The mutant only dimerizes upon VEGF$_{165}$ binding ($D_{confocal}$ = 0.017 ± 0.007 μm² s⁻¹). In summary, our data indicate that VEGFR1 remains predominantly an inactive monomer in the unligated state. Perhaps the polarized electrostatic surface of the ECD (D4-D7) prevents receptor dimerization in the absence of ligand[24,65,67], and mutation of C471 does not induce autophosphorylation of the receptor. Therefore, we speculate that removing ECD inhibition might spontaneously activate the VEGFR1, as observed in many RTKs[73–76] and is often linked to pathological manifestations[77].

To test this, we measured the autophosphorylation of Y1213 and Y1175 in the ECD-deleted (ΔECD) construct of VEGFR1 and VEGFR2, respectively (Fig. 3f, g and Supplementary Fig. 3f, g). As shown previously[76], the VEGFR2 ΔECD construct was constitutively activated (Supplementary Fig. 3f) and linearly phosphorylates Y1175 in the single-cell assay (Fig. 3f). Counter-intuitively, the deletion of ECD did not activate the VEGFR1 even at the higher receptor concentration (Fig. 3g and Supplementary Fig. 3g). We speculate that the TM-JM segment connecting the ECD and kinase domain (Fig. 1) may be constitutively inhibiting the ligand-independent activation of VEGFR1.

### The transmembrane domain does not drive ligand-independent VEGFR1 activation

The TM segment is a major driving force for RTK dimerization. The dynamic equilibrium between receptor dimer and monomer is rotationally coupled to the orientation of the TM segment[78,79]. VEGFR2 TM segment adopts two dimer structures, ligand-independent and ligand-dependent (Fig. 4d)[26,80]. The sequence comparison between the VEGFR1 and VEGFR2 shows that the residues at the ligand-independent

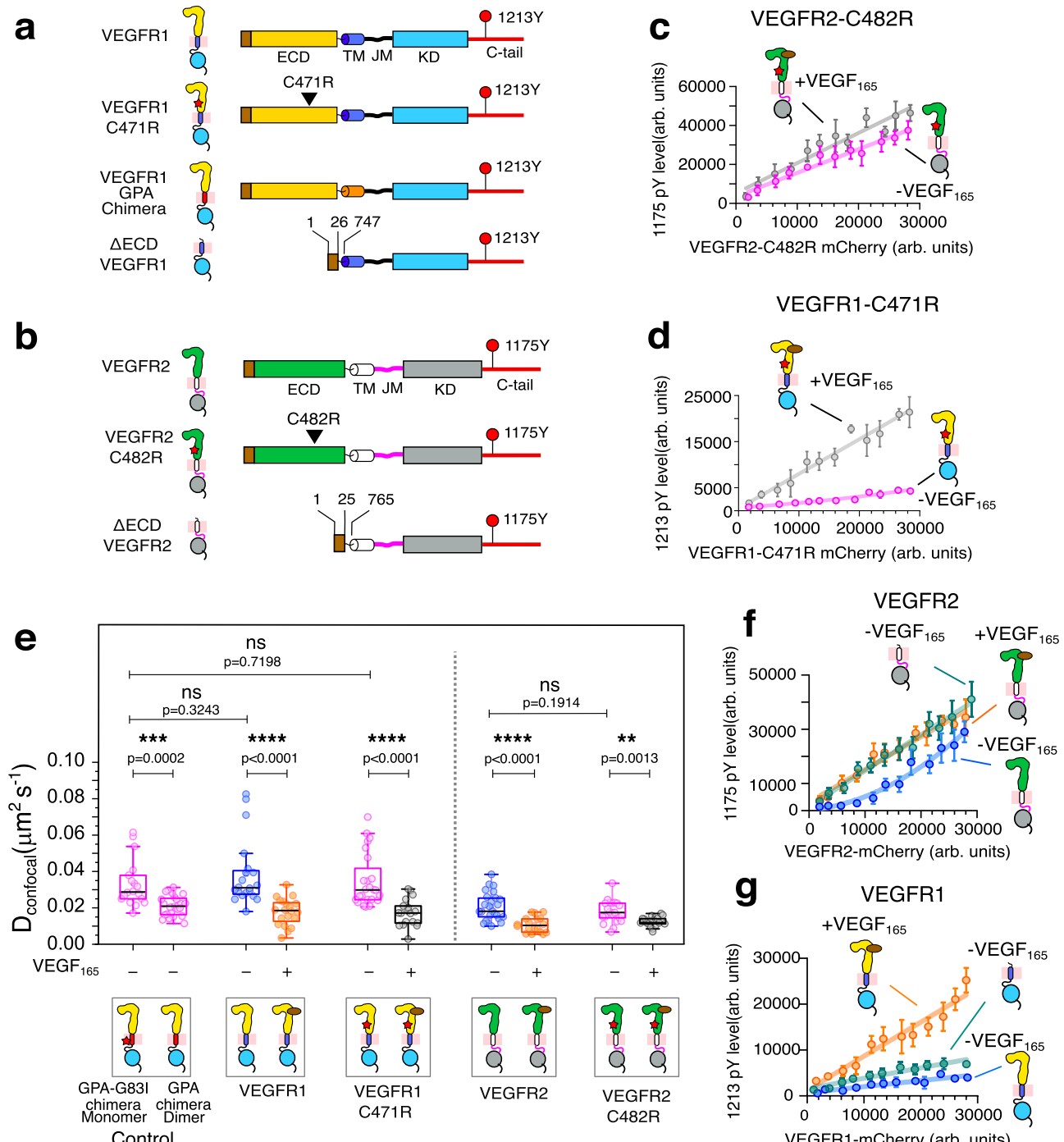

**Fig. 3 | Probing the role of ECD in stabilizing the VEGFR1 autoinhibited state.**
**a**, **b** Schematic representation of VEGFR1 (**a**) and VEGFR2 (**b**) constructs used in this study. **c** The plot of Y1175 phosphorylation level against the expression level of the constitutively activated C482R mutant of VEGFR2 in the presence or absence of VEGF$_{165}$. [$n = 73$ (−VEGF$_{165}$) and 75 (+VEGF$_{165}$) cells examined over 5 independent experiments. Data are presented as mean values ± SD]. **d** The plot of Y1213 phosphorylation versus VEGFR1-C471R expression in the presence and absence of ligand. [$n = 70$ (−VEGF$_{165}$) and 80 (+VEGF$_{165}$) cells examined over five independent experiments. Data are presented as mean values ± SD]. **e** The diffusion coefficient measured from FRAP studies of indicated constructs of VEGFR1 and VEGFR2 in the presence and absence of VEGF$_{165}$ are plotted. VEGFR1-GPA chimera and VEGFR1-GPA-G83I chimera represent dimer and monomer controls, respectively. Each data point in the box plot reflects the diffusion coefficient of the selected cell, and the black line indicates the mean value. n = 18 (VEGFR1-TM$^{gPA-G83I}$), 24 (VEGFR1-TM$^{gPA}$), 20 (VEGFR1-VEGF$_{165}$), 20 (VEGFR1 + VEGF$_{165}$) 24 (C471R-VEGF$_{165}$), 18

(C471R + VEGF$_{165}$), 25 (VEGFR2-VEGF$_{165}$), 18 (VEGFR2 + VEGF$_{165}$) 21 (C482R-VEGF$_{165}$), and 18 (C482R + VEGF$_{165}$) cells examined over 8 independent experiments. An unpaired two-tailed $t$-test was used to calculate significance. Boxplots represent quartiles. The data points outside the whisker range are set as outliers. The black line inside the box represents the median value. **f, g** The plot of the phosphorylation level of Y1175 in VEGFR2 (**f**) and Y1213 in VEGFR1 (**g**) against the indicated receptor expression level in the presence and absence of the ligand. In panel f, n = 72 (VEGFR1-VEGF$_{165}$), 70 (VEGFR1 + VEGF$_{165}$), and 74 (ΔECD-VEGFR1) cells were examined over four independent experiments. In (**g**), $n = 70$ (VEGFR2-VEGF165), 62 (VEGFR2 + VEGF165), and 91 (ΔECD-VEGFR2) cells were examined over four independent experiments. Data points are represented as mean values ± SD. All data were plotted using GraphPad Prism Ver 9.5.1. The boxplots were generated using Origin Pro 2020b. All the schematics and icons were designed using Inkscape Ver1.2. Source data are provided as a Source Data file for (**b**–**g**). See Supplementary Figs. 3, 4.

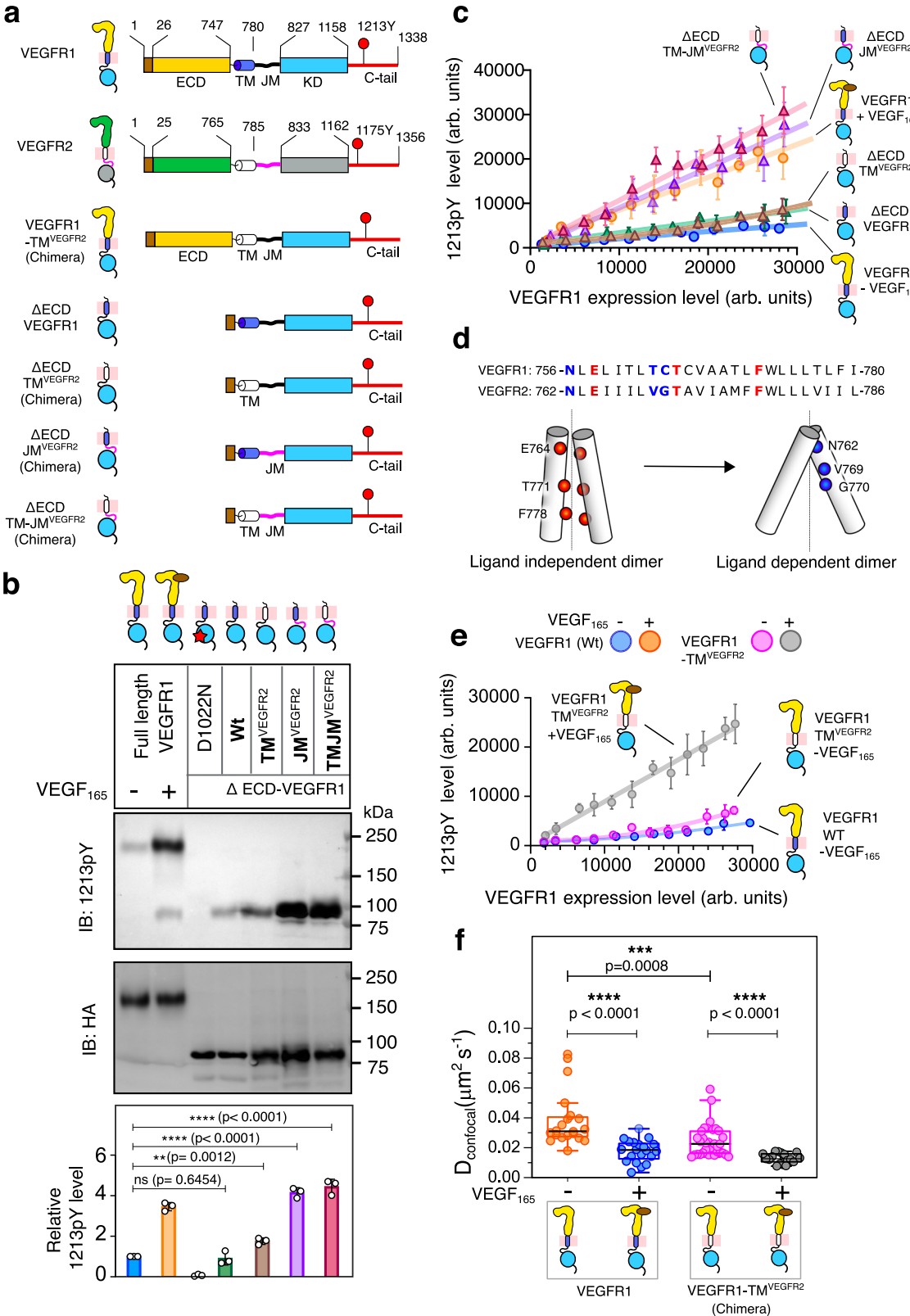

dimer interface are conserved (Fig. 4d). In contrast, the residues at the ligand-dependent dimer interface are not conserved. We speculate that the presence of T763 and C764 may bias the TM structure to the ligand-dependent dimer over the ligand-independent dimer[81,82]. Thus, in the absence of a ligand, structural incompatibility between the active TM dimer and the unligated ECD dimer may render the VEGFR1 to remain in the monomeric state[83]. Ligand binding favors the TM

structure towards a ligand-dependent dimer. To test that, we mutated the T761, T763, or C764 individually in VEGFR1 to the corresponding residue in VEGFR2 and measured the Y1213 phosphorylation (Supplementary Fig. 5c). We also replaced the TM segment of VEGFR1 with VEGFR2 in the full-length and ΔECD construct of VEGFR1 (Fig. 4a–c, e, and Supplementary Fig. 5c). We observed that none of the single-mutant and TM chimeric constructs could activate the VEGFR1 ligand

**Fig. 4 | Functional analysis of TM and JM segments in ligand-independent activation of VEGFR1. a** Schematic representation of VEGFR1 and VEGFR2 constructs used in this study. **b** Immunoblot showing the phosphorylation of Y1213 in the indicated constructs of VEGFR1. The expression level of the VEGFR1 is determined using an antiHA antibody. The bar plot in the lower panel represents the relative Y1213 phosphorylation level determined from densitometric analysis. Data are presented as mean values ± SD from three independent experiments. An unpaired two-tailed t-test is used to calculate the significance. **c** The plot of the Y1213 phosphorylation level against the expression level of VEGFR1 ΔECD and wt from the single-cell assay. $n = 48$ (VEGFR1-VEGF$_{165}$), 69 (VEGFR1 + VEGF$_{165}$), 67 (ΔECD-VEGFR1), 75(ΔECD-VEGFR1$^{TM}$) 91 (ΔECD-VEGFR1$^{JM}$), and 95 (ΔECD-VEGFR1$^{TMJM}$) cells were examined over four independent experiments. Data points are presented as mean values ± SD. **d** Sequence alignment of TM segment of VEGFR1 and VEGFR2. The amino acid residues at the VEGFR2 ligand-independent and dependent dimer interface are colored red and blue, respectively. Below, is the cartoon of the ligand-independent and dependent VEGFR2 TM dimer[26]. **e** The plot of Y1213 phosphorylation versus the expression level of indicated VEGFR1 constructs. $n = 63$(VEGFR1-VEGF$_{165}$) and 73(VEGFR1TM$^{VEGFR2}$-VEGF$_{165}$), 80(VEGFR1TM$^{VEGFR2}$ + VEGF$_{165}$) cells examined over 5 independent experiments. Data are presented as mean values ± SD. **f** The dimerization propensity of indicated VEGFR1 constructs is probed from the diffusion coefficient measured by the FRAP experiment. $n = 20$(VEGFR1-VEGF$_{165}$), n = 20(VEGFR1 + VEGF$_{165}$), 33(VEGFR1TM$^{VEGFR2}$-VEGF$_{165}$), 17(VEGFR1TM$^{VEGFR2}$ + VEGF$_{165}$) cells examined over 10 independent experiments. An unpaired two-tailed t-test was used to calculate significance. Boxplots represent quartiles, and whiskers correspond to range. The data points outside the whisker range are set as outliers. The black line in the box represents the median value. All data were plotted using GraphPad Prism Ver 9.5.1. The boxplots were generated using Origin Pro 2020b. All the schematics and icons were designed using Inkscape Ver1.2. Source data are provided as a Source Data file for the panels (**b**, **c**, **e**, **f**). See Supplementary Fig. 5.

independently. The FRAP analysis suggests that the full-length VEGFR1-TM$^{VEGFR2}$ chimera has a higher propensity to form a ligand-independent dimer ($D_{confocal} = 0.022 ± 0.007\ \mu m^2\ s^{-1}$) than the wild-type VEGFR1 (Fig. 4f, Supplementary Fig. 5e, f, and Supplementary Table 3). Suggesting the TM segment of VEGFR1 is a weak dimerization motif compared to VEGFR2 in a ligand-free state. To determine if a stronger TM dimerization motif could spontaneously phosphorylate Y1213 independent of VEGF$_{165}$ stimulation, we turned to the VEGFR1-TM$^{GPA}$ chimera (Supplementary Fig. 4i). We observed that even the VEGFR1-TM$^{GPA}$ chimera could not phosphorylate the Y1213 constitutively. The inability of the VEGFR1-TM$^{VEGFR2}$ or VEGFR1-TM$^{GPA}$ dimer to activate the kinase-domain ligand independently is counterintuitive. These data indicate that the regulatory elements downstream of the TM segment may constitutively autoinhibit VEGFR1 in the unligated state. Therefore, we replaced the JM or TM-JM segment of VEGFR1 with the VEGFR2 in the ΔECD background. Replacing the JM segment spontaneously activates the kinase and linearly phosphorylates Y1213 (Fig. 4b, c and Supplementary Fig. 5d). Together, we conclude that the JM segment is a key regulator of VEGFR1 activation.

## The electrostatic latch stabilizes the inactive conformation of the JM segment

A repressor sequence present in the JM segment of VEGFR1 is known to inhibit the downstream signaling and cell migration constitutively[30]. The JM segment of VEGFR1 and VEGFR2 are homologous and have minor differences in the amino acid sequence (Fig. 5a). In the auto-inhibited state, the kinase domain of VEGFR1 (PDB ID: 3HNG) and VEGFR2 (PDB ID: 4AGC)[21] adopts a JM-in-like inactive conformation, found in the PDGFR family of kinases (Supplementary Fig. 6a)[19–21]. The JM-B segment is buried deep into the catalytic site, stabilizing the folded conformation of the activation loop and preventing the rearrangement of the N- and C-lobe to an active state. The conformation of the JM-Z region sets the direction of the rest of the JM segment inward to the kinase domain. In spite of sharing a high degree of sequence and structural homology, it is not clear how the JM segment of VEGFR1 is differentially regulated from VEGFR2.

To find answers, we revisited the structure of the VEGFR and PDGFR family of kinases. In the crystal structure of VEGFR, the JM-S segment was unresolved. We model the JM-S segment of VEGFR1 and VEGFR2 based on the inactive structure of PDGFR (PDB ID: 5K5X)[84] (Supplementary Fig. 6a). The structural evaluation revealed two key aspects: First, the JM-S segment carries an overall negative charge. Second, the C-lobe of VEGFR1 has a positive charge patch, which is absent in the other VEGFR and PDGFR family (Fig. 5a–c). In our model, the positive charge residues (K1142, K1079, or R1146) in the C-lobe of the VEGFR1 kinase domain form salt bridges with the negatively charged residues (D802 and E803) in the JM-S segment (Fig. 5b). In VEGFR2, the corresponding residues in the C-lobe do not form salt bridges with the JM-S segment (Fig. 5c and Supplementary Fig. 6g). We

hypothesized that the unique salt bridge between the JM-S and the C-lobe acts as an electrostatic latch that stabilizes the JM-in conformation of VEGFR1, rendering it constitutively inactive in the ligand-free state.

To test our hypothesis, we use molecular dynamics simulation to find the relative stability of the electrostatic latch in VEGFR1 and VEGFR2, respectively (Supplementary Fig. 6f, g). Our analysis of the distance between the ion pairs suggests that the electrostatic latch in VEGFR1 may be more stable than in VEGFR2 (Supplementary Fig. 6f–i). We observed that the E803 and D802 in the JM-S segment of VEGFR1 maintained electrostatic contact with the respective C-lobe residues during the simulation. We speculate that the electrostatic latch may be an integral component of the autoinhibited VEGFR1 structure and may regulate the transition between an inactive to an active conformation.

## Removing JM inhibition increases the ligand-independent activation of VEGFR1

To evaluate the structure and function of the electrostatic latch, we interrogate two VEGFR1 constructs, VEGFR1-JM$^{VEGFR2}$ chimera, and triple mutant (3 M) (where positively charged residues in the C-lobe K1142S, K1079Q, and R1146T are mutated to the corresponding residues in VEGFR2) (Figs. 5d, e, 6a). Our structural model shows that the electrostatic latch is broken in the VEGFR1-JM$^{VEGFR2}$ and 3 M construct (Fig. 5d, e and Supplementary Fig. 6h–i). We speculate that perturbing the electrostatic latch may destabilize the autoinhibitory interaction of the JM-B. Thus, replacing the VEGFR1 JM segment with VEGFR2 or the triple mutant (3 M) may restore ligand-independent activation of VEGFR1. In the single-cell assay, replacing VEGFR1 JM or TM-JM segments with VEGFR2 (Fig. 6b, c) restores the concentration-dependent autophosphorylation in the ligand-free state. The 3 M mutant partially restored the Y1213 phosphorylation, suggesting a critical role for the electrostatic latch in stabilizing the inactive JM-in structure (Fig. 6d). However, the complete restoration of the ligand-independent VEGFR1 autophosphorylation might require additional JM restraint to be removed.

To investigate why perturbing the electrostatic latch does not fully restore the ligand-independent activation of VEGFR1, we revisited the JM-in structure of VEGFR2. We observed that the conserved Y801 in the JM-B region of VEGFR2, which forms an H-bond with the critical glutamic acid residue in C-helix in the PDGFR[19,85], is moved out of the catalytic site and does not interact with the C-helix (Supplementary Fig. 7a, b). The JM-B segment of VEGFR1, which is shorter by one residue than VEGFR2, is unresolved in the crystal structure (PDB ID 3HNG) (Fig. 5a). Based on structural modeling, we predict that moving the corresponding Y794 in VEGFR1 to (−)1 position may place the Y794 in the catalytic site, allowing it to interact with the glutamic acid residue in C-helix (Supplementary Fig. 7a, b). We hypothesize that if the Y794 is moved out of the catalytic site and simultaneously removing the electrostatic latch may activate the VEGFR1 ligand independently.

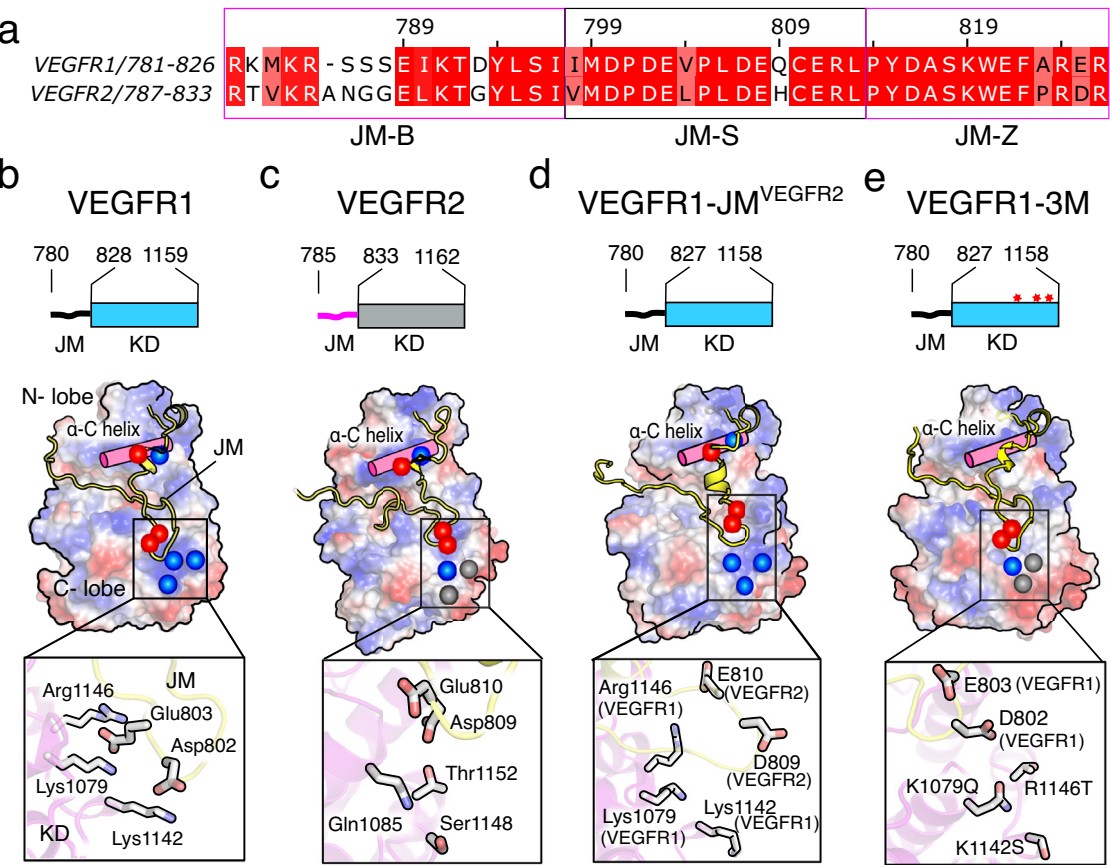

**Fig. 5 | Structural analysis of JM inhibition in VEGFR1. a** Sequence alignment of VEGFR1 and VEGFR2 JM segments. The domain boundaries are defined based on FMS-like tyrosine kinase 3 structure (PDB ID: 1RJB)[20]. **b**−**e** The upper panel shows the schematic diagram of the JM-KD construct used for MD simulation. The bottom panel is the space-filled model of VEGFR1 and VEGFR2 constructs. The electrostatic surface potentials are colored blue and red for the positive and negatively charged sidechains, respectively. The polar uncharged residues are colored grey. The electrostatic interactions (electrostatic latch) between JM and the C-lobe of the kinase domain are shown in the inset. The space-filled model was generated by using PyMOL Molecular Graphics System, Version 2.5.2 Schrödinger, LLC. See Supplementary Fig. 6.

Using the single-cell assay, we determined the ligand-independent activation of VEGFR1 in a ΔS mutant (where three consecutive serine residues, 786-788 at the JM-B, are removed) and a ΔS mutant in the 3 M background (Fig. 6a). We observed that the ΔS mutant could partially restore the Y1213 phosphorylation without ligand (Fig. 6d). To our delight, the ΔS mutant introduced in the 3 M background restored the ligand-independent VEGFR1 activation to a level comparable to the VEGFR1-JM$^{VEGFR2}$ chimera (Fig. 6d and Supplementary Fig. 7c). We ask if removing the JM inhibition is enough to remodel the phosphorylation kinetics of VEGFR1.

**Removing JM inhibition remodels transient phosphorylation of VEGFR1 to sustained phosphorylation**

The FRAP experiment shows an increased dimeric propensity for the VEGFR1-JM$^{VEGFR2}$ ($D_{confocal} = 0.0205 \pm 0.0059\ \mu m^2\,s^{-1}$) and VEGFR1-TMJM$^{VEGFR2}$ ($D_{confocal} = 0.0202 \pm 0.007\ \mu m^2\,s^{-1}$) chimera (Fig. 6e, Supplementary Fig. 7d-e, and Supplementary Table 3). Suggesting the JM-in conformation of VEGFR1 is incompatible with the ligand-independent TM dimer. Stimulating VEGFR1-JM$^{VEGFR2}$ ($D_{confocal} = 0.0127 \pm 0.0047\ \mu m^2\,s^{-1}$) and VEGFR1-TMJM$^{VEGFR2}$ ($D_{confocal} = 0.0131 \pm 0.005\ \mu m^2\,s^{-1}$) with VEGF$_{165}$ induces the receptor oligomerization and also increases the relative fraction of Y1213 phosphorylation compared to wild-type VEGFR1 (Fig. 6e, f and Supplementary Table 3). Removing the JM inhibition in the VEGFR1-TMJM$^{VEGFR2}$ chimera increases the rate of Y1213 phosphorylation ($0.16 \pm 0.03$ arb. units/min). It also remodels the phosphorylation half-life from transient to sustained ($t_{1/2} = 48 \pm 13.8$ min) (Fig. 6g, h,

Supplementary Fig. 7f−h, and Supplementary Table 4). We conclude that the multiple interactions between the JM segment and the kinase core are the dominant force in autoinhibiting the VEGFR1 constitutively in the unligated state.

## Discussion

To summarize, we presented a molecular mechanism explaining how VEGFR1 and VEGFR2 are differentially autophosphorylated in the unligated state and upon ligand stimulation (Fig. 7). An equilibrium shift model of multiple species including a monomeric receptor and various dimers, previously proposed for EGFR, explains the ligand-independent activation of VEGFR2[58,59,86]. The ECD of EGFR in the unligated state forms a head-to-head dimer that keeps the TM segment separated, preventing the two kinase domains from adopting an asymmetric active dimer[87–89]. Consistent with the previous finding, our results show that the TM segment in VEGFR2 dimerizes in the absence of a ligand, allowing the two kinase domains to transphosphorylate at a critical receptor concentration[26,54]. In contrast, we found that VEGFR1 remains predominantly as a monomer in the absence of a ligand, and replacing the TM segment of VEGFR1 with VEGFR2 allows the receptor to dimerize independently of the ligand (Fig. 4f). However, forced dimerization of the VEGFR1 failed to activate the kinase domain (Fig. 4e and Supplementary Fig. 4i).

Unlike other RTKs[59,76], removal of the ECD does not spontaneously activate the VEGFR1 (Fig. 4c). Replacing the JM segment of the VEGFR1 with that of VEGFR2 activates the kinase domain ligand independently (Fig. 6c). Our data supports a central role for the JM segment

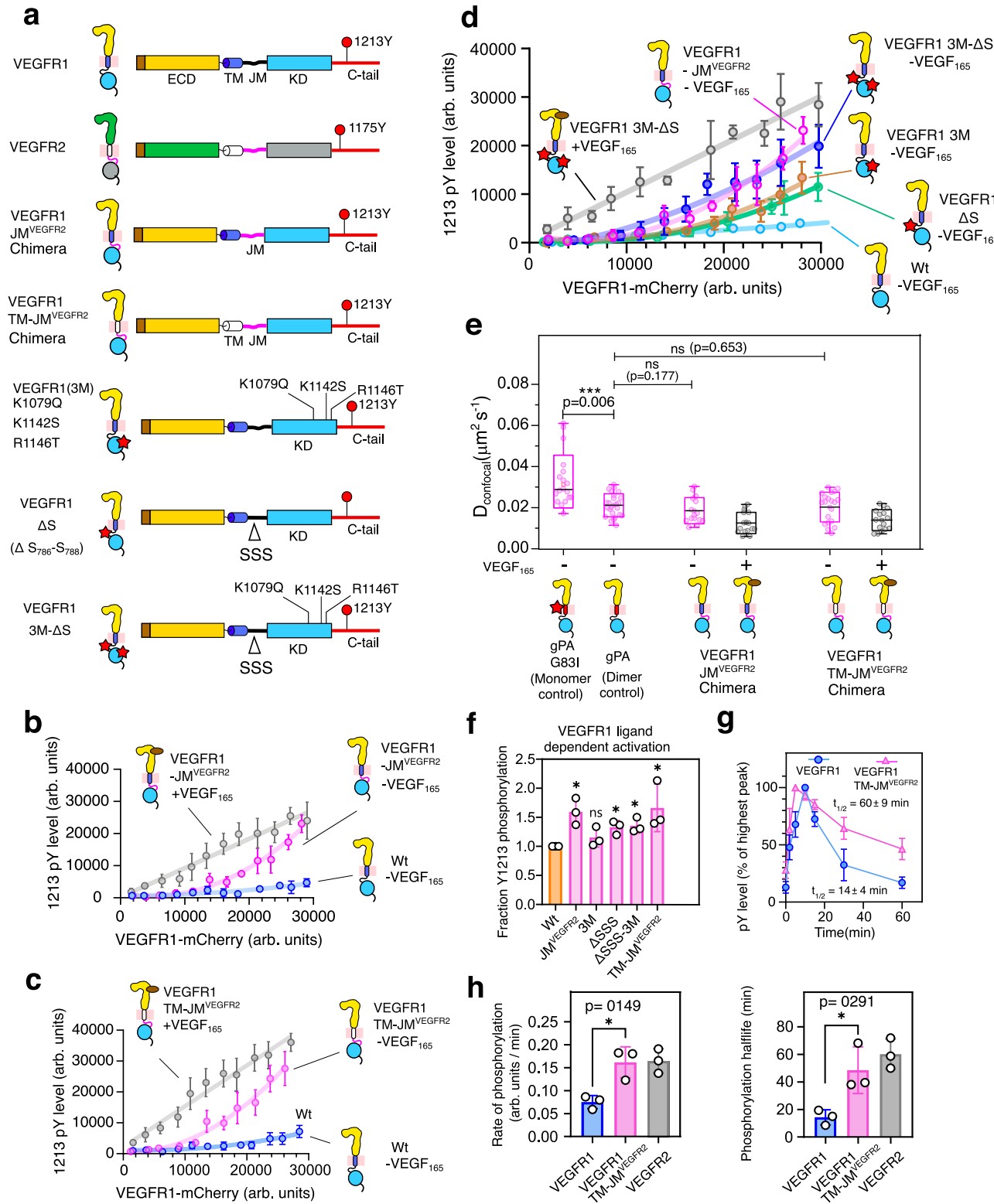

in balancing the equilibrium between an inactive and active state of VEGFR1. In the unligated state, the electrostatic latch and H-bond interaction steaming from Y794 in JM-B may shift the equilibrium towards the inactive state, thus making VEGFR1 an inefficient kinase. The JM-in conformation of VEGFR1 suppresses the spontaneous kinase activation (Figs. 5 and 6)[26]. We therefore proposed that slow kinase activation along with the cellular protein tyrosine (PTP) activity[90,91] maintains the basal activity of VEGFR1 constitutively inhibited (Fig. 7). The role of the JM segment of VEGFR is in contrast to the EGFR, where

the JM segment of the receiver kinase stabilizes the active asymmetric dimer by interacting with the C-lobe of the activator and couples to the TM dimer[19,59,89,92].

The asymmetric dimer of the EGFR kinase domain generates asymmetry in the phosphorylation of tyrosine residues in the C-terminal tail, with a clear preference for activator[93]. A possible asymmetry in the phosphorylation of tyrosine residues in the C-terminal tail may occur in an asymmetric heterodimer of VEGFR1 and VEGFR2[25]. Our data suggests that the heterodimer of two kinetically

**Fig. 6 | Functional study of JM segment in regulating concentration-dependent activation of VEGFR1. a** Schematic representation of chimeric constructs and mutants of VEGFR1 used in this study. **b, c** Concentration-dependent activation of VEGFR1 constructs is determined using a single-cell assay in the presence and absence of ligands. In panel (**b**), $n = 51$ (VEGFR1-VEGF$_{165}$), 107 (VEGFR1-JM$^{VEGFR2}$-VEGF$_{165}$) and 86 (VEGFR1-JM$^{VEGFR2}$ + VEGF$_{165}$) cells examined over six independent experiments. In panel c, n = 55 (VEGFR1-VEGF$_{165}$), 95 (VEGFR1-TMJM$^{VEGFR2}$-VEGF$_{165}$), and 84 (VEGFR1-TMJM$^{VEGFR2}$ + VEGF$_{165}$) cells examined over six independent experiments. Data are presented as mean values ± SD. **d** The plot of Y1213 phosphorylation versus the expression level of indicated VEGFR1 constructs. $n = 56$ (VEGFR1-VEGF$_{165}$), 84 (VEGFR1 3 M -VEGF$_{165}$), and 70 (VEGFR1 ΔSSS - VEGF$_{165}$), 83 (VEGFR1 3 M ΔSSS - VEGF$_{165}$), 107 (VEGFR1-JM$^{VEGFR2}$ - VEGF$_{165}$) and 65 (VEGFR1 3 M ΔSSS + VEGF$_{165}$) cells examined over six independent experiments. Data are presented as mean values ± SD. **e** The dimerization propensity of the indicated VEGFR1 construct is probed from the diffusion coefficient measured using the FRAP experiment. $n = 18$ (VEGFR1-TM$^{gPA-G83I}$), 24 (VEGFR1-TM$^{gPA}$), 23 (VEGFR1-JM$^{VEGFR2}$-VEGF$_{165}$), 18 (VEGFR1-JM$^{VEGFR2}$ + VEGF$_{165}$) 20 (VEGFR1-TMJM$^{VEGFR2}$ - VEGF$_{165}$) and 17 (VEGFR1-TMJM$^{VEGFR2}$ + VEGF$_{165}$) cells examined over eight independent experiments. An unpaired two-tailed t-test was used to calculate significance. Boxplots represent quartiles, and whiskers correspond to range. The lower whisker shows the 5th percentile, and the upper whisker shows the 95th percentile. The black line in the box represents the mean value. **f** The relative fraction of phosphorylated Y1213 for the indicated VEGFR1 construct upon VEGF$_{165}$ stimulation is shown as a

bar diagram. The fraction phosphorylated was obtained from the slope of the ligand-dependent activation of the respective VEGFR1 construct, as described in panels (**b, c, d**), and normalized against the wt data. Data are presented as mean values ± SD from three independent experiments. An unpaired two-tailed t-test was used to calculate significance. (VEGFR1-JM$^{VEGFR2}$ *$p$ = 0.0108, VEGFR1 3 M $^{ns}p$ = 0.1827, VEGFR1 ΔSSS *$p$ = 0.0101, VEGFR1 3 M ΔSSS *$p$ = 0.0172, VEGFR1-TMJM$^{VEGFR2}$ *$p$ = 0.0286). **g** The densitometric analysis of the Y1213 phosphorylation level at the indicated time points for the wt (blue) and chimeric construct (magenta) of VEGFR1 (Fig. 2g and Supplementary Fig. 7f). The phosphorylation level at each time point is normalized against the highest intensity observed for the respective data set. The error bar represents the standard deviation of three independent experiments. Data are presented as mean values ± SD. **h** The rate of phosphorylation (left panel) and phosphorylation $t_{1/2}$ (right panel) of the indicated VEGFR constructs are determined from the densitometric analysis of ligand-dependent activation, as described in Fig. 2g, h and Fig. 6g and Supplementary Fig. 7f. The error bar represents the standard deviation of three independent experiments. Data are presented as mean values ± SD from three independent experiments. An unpaired two-tailed t-test was used to calculate the significance. All the data were plotted using GraphPad Prism Ver 9.5.1. The boxplots were generated using Origin Pro 2020b. All the schematics and icons were designed using Inkscape Ver1.2. Source data are provided as a Source Data file for panels b-h. See Supplementary Figs. 2, 7.

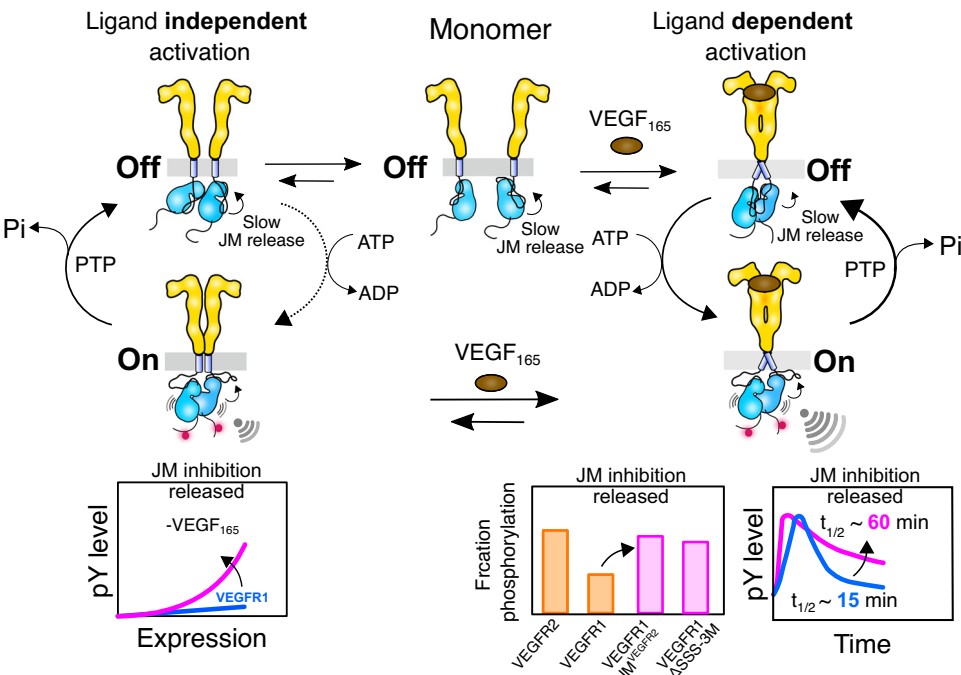

**Fig. 7 | The proposed model for VEGFR1 autoinhibition.** Right: The ligand binding to the ECD induces receptor dimerization and rearranges the TM-JM segment. Slow release of JM inhibition in VEGFR1 leads to transient tyrosine phosphorylation at the C-terminal tail. Faster release of JM inhibition in VEGFR1 chimera and the mutants remodels the tyrosine phosphorylation to be sustained. Left: Ligand-independent activation of VEGFR1 is suppressed due to a delicate balance between the slow release of JM inhibition and protein tyrosine phosphatase (PTP) activity[90]. The schematics and icons were designed using Inkscape Ver1.2.

different kinases may generate distinctly different tyrosine phosphorylation patterns compared to the homodimer, hence generating an altered signaling output[25,94,95].

The ligand bias by the ECD dimer generates differential signaling output in EGFR[96,97] and is a crucial determinant for deciding the cell fate[98]. A transient versus sustained Erk activation generated by stimulating EGFR with two different ligands, EGF or NGF, switches the signaling outcomes to differentiation from the proliferation[99,100]. Recent studies suggest that the subtle structural difference in the

ligand-induced ECD dimer may determine the differential signaling output in EGFR[96,97]. Here, we studied how a common ligand induces different outputs in two VEGF receptor isomers. We proposed that the slow removal of JM inhibition in VEGFR1 may generate transient phosphorylation of tyrosine residues (Fig. 7). We speculate that the bias signaling by the two VEGF receptor isomers may account for the differential response of VEGFR1 in promoting macrophage proliferation[101], whereas VEGFR2 supports differentiation of endothelial cell (EC) progenitors[102].

## Methods

### DNA constructs

The cDNA encoding wild-type human VEGFR1 (Residue number 1–1338; pDONR223-FLT1 was a gift from William Hahn & David Root; Addgene plasmid # 23912) was subcloned into pcDNA-mCherry vector between the HindIII and Kpn1 restriction sites. The plasmid encoding human wild-type VEGFR2 tagged with mCherry (cloned in pBE-vector; 108854) was a gift from Kalina Hristova (Johns Hopkins University, Baltimore, MD). All point mutations and deletion constructs were prepared by PCR-based mutagenesis strategy[103]. For the VEGFR1 ECD deletion constructs, Haemagglutinin (HA) tag was incorporated at the 5´ ends. The list of primers used is tabulated in Table S3. In brief, PCR-amplified product at the expected molecular weight was extracted and purified from 0.6% agarose gel and incubated with Dpn1 at 37 °C for 3 h. The Dpn1 digested PCR product was further purified from the reaction mixture and incubated with Polynucleotide Kinase 4 (PNK4) and Ligase in Tris buffer containing 10 mM MgCl$_2$ and 2 mM ATP. Finally, the ligated mixture was transformed into the *E.coli* Top10 cell. Chimeric constructs were prepared using the Gibson assembly approach[104]. All the chimeric constructs were cloned into BamH1 and Xho1 restriction sites in a pcDNA-mCherry vector (mCherry2-N1 was a gift from Michael Davidson, Addgene Plasmid # 54517)

### Cell culture, immunofluorescence

Chinese Hamster Ovary (CHO) (obtained from National Center for Cell Science- India), African green monkey kidney fibroblast-like cell line (COS-7) (obtained from National Center for Cell Science- India), and murine macrophage (RAW264.7) cell lines (obtained from ATCC, Catalog No. TIB-71) were cultured in DMEM supplemented with 10% FBS, 50 µg/ml penicillin, and streptomycin at 37 °C with 5% CO$_2$. In the single-cell experiments and immunoblotting, VEGFR1 or VEGFR2 was transiently transfected in respective cell lines. In brief, the COS-7 or the CHO cells were grown on coverslips (10 mm, #1 thickness) up to a confluency of 80% and transfected with Lipofectamine 2000 in serum-free media (Opti-MEM). After 6 hours of incubation, cells were supplemented with complete media containing 10% FBS and allowed to grow for another 12 hours. Alternatively, RAW264.7 cells were nucleofected using the D032 program in the Lonza nucleofector. $4 \times 10^7$ cells were incubated with 4 µg of DNA in Amaxa buffer for 5 min at room temperature. Nucleofected cells were resuspended in DMEM supplemented with 15% FBS. 18 h post-transfection, cells were serum-starved by growing in serum-free opti-MEM for another 8 h. To activate VEGFR1 or VEGFR2, transfected cells were treated with 100 nM VEGF$_{165}$ for 5 mins, then immediately washed with ice-cold PBS two times and fixed by treating with 4% paraformaldehyde (PFA) in PBS for 30 min at room temperature. After fixation, cells were washed five times with 1× PBS and permeabilized with 0.2% PBST (1X PBS and 0.2% Triton-X 100) for 5 min at room temperature. Cells were blocked with 1% BSA for 1 h, followed by staining with anti-phosphotyrosine antibodies for VEGFR2 or VEGFR1 overnight at 4 °C (dilution 1:200 for anti phospho-VEGFR1 and anti phospho-VEGFR2 antibody). After overnight incubation, coverslips were washed three times with PBST, followed by incubation with a secondary antibody conjugated with FITC for 2 h at room temperature. Finally, coverslips were mounted with prolonged gold. in between each step, the coverslips were washed three times with PBST and finally with PBS before mounting. All antibodies used, along with the catalog numbers and dilutions, are mentioned in Supplementary Table 1.

### Microscopy and image analysis

Transiently transfected COS-7 cells were imaged by an Olympus IX81 epifluorescence microscope. For CHO and RAW264.7 cell line, all images were acquired with the Leica SP8 confocal platform using an oil immersion HC PL APO pinhole was set at 1 AU. Image scanning was done in bidirectional mode at 500 Hz. Expression of mCherry fused VEGFR constructs and the phosphorylation levels was imaged using 40 mW/552 nm and 20 mW/488 nm solid-state lasers, respectively. For mCherry, $\lambda_{ex}$ and $\lambda_{em}$ were set to 552 nm and 576–651 nm, respectively. For FITC, $\lambda_{ex}$ and $\lambda_{em}$ were set to 488 nm and 505-531 nm, respectively. Laser power was set at 0.08 mW and 0.04 mW for mCherry and FITC channels, respectively. The background fluorescence was corrected by subtracting the respective fluorescence measured for untransfected cells. Image analysis was done by selecting ROIs drawn manually using ImageJ freehand tool[105], as explained previously[59]. Two sets of ROI were selected; ROI1 was drawn to calculate total cell intensity, and ROI2 was drawn at the peripheral region of the cell to calculate cytoplasmic intensity (Supplementary Fig. 1d). The fluorescence intensity of mCherry or FITC at the membrane ($I_m$) was calculated by:

$$I_m = I_{ROI1} - I_{ROI2}$$ where $I_{ROI1}$ and $I_{ROI2}$ are fluorescence of mCherry or FITC at the respective ROI.

To analyze the tyrosine phosphorylation as a function of receptor expression level, the intensity of the mCherry channel was binned in the range of 2500 arb. units, and the average value of mCherry intensity was plotted against the corresponding average value of FITC intensity. For measuring the normalized phosphorylation level, the intensity from the FITC channel was normalized by the corresponding mCherry intensity for the respective bin, the cells were binned at an intensity range of 10000 arb. units.

### Immunoblotting

Activation of VEGFR1 or VEGFR2 was performed as mentioned in the above section. Briefly, CHO cells were cultured in DMEM and transfected with VEGFR1 and VEGFR2 constructs. 18 h post transfection, cells were serum-starved by growing in serum-free opti-MEM for another 8 h and activated with 100 nM VEGF$_{165}$ for 5 mins. After activation, cells were washed with ice-cold PBS before being incubated in RIPA buffer (10 mM Tris-Cl, pH 8.0, 140 mM NaCl, 1 mM EDTA, 0.1% SDS, and 1% Triton X-100) containing protease and phosphatase inhibitors (2 mM Benzamidine, 1 mM PMSF), and then sonicated on ice. Protein samples were prepared by heating them with 5X loading buffer, resolving them on a 6% SDS-PAGE, and blotted onto a PVDF membrane. The blot was blocked with 5% skimmed milk in 1× TBS and 0.1% TWEEN-20 for 1 h at RT and then incubated with primary antibody at 4 °C overnight (dilution 1:1000 for anti VEGFR2, anti Phospho-VEGF Receptor 2, anti VEGFR1 and anti Phospho-VEGFR1 Antibody; 1:2000 for anti HA antibody). The unbound primary antibody was removed by washing the blot three times with 1× TBST (0.1% tween-20), followed by incubation with a secondary antibody diluted in 3% skimmed milk (dilution 1: 2000 for rabbit and goat HRP secondary antibody; 1:3000 for mouse HRP secondary antibody). Blot was washed three times with 1× TBST and two times with 1XTBS wash before developing with the Clarity™ Western ECL substrate kit (Bio-Rad). The images were acquired using the Bio-Rad Chemidoc system. The densitometric analysis was performed using ImageJ. All antibodies used, along with the catalog numbers and dilutions, are mentioned in Supplementary Table 1.

### Cell-based tyrosine phosphorylation kinetic measurements

Chinese Hamster Ovary (CHO) cell line was cultured in DMEM and transfected with VEGFR constructs. 18 h post transfection, cells were serum-starved in Opti-MEM for another 8 h. The VEGFR tyrosine phosphorylation was stimulated by treating with VEGF$_{165}$. Samples were collected at the indicated time points and analyzed by immunoblotting (dilution 1:1000 for Anti phosphotyrosine Antibody, anti VEGFR2, anti Phospho-VEGF Receptor 2, anti VEGFR1 and anti Phospho-VEGFR1 Antibody; 1:2000 for anti HA antibody). Densitometry analysis of immunoblots was performed using ImageJ software. All antibodies used, along with the catalog numbers and dilutions, are mentioned in Supplementary Table 1.

## FRAP experiment

CHO cells were seeded in a 35 mm glass-bottomed Petri dish at a density of $0.5 \times 10^6$ cells per plate. 24 h post seeding, cells were transfected with VEGFR1 or VEGFR2 constructs. The cells were grown for 18 h post-transfection and were serum-starved for 6 h before ligand stimulation. Before activating with $VEGF_{165}$, cells were washed with pre-warmed PBS(1×) and activated with 100 nM $VEGF_{165}$ in Opti-MEM.

FRAP experiments were conducted using a Leica SP8 confocal microscope at room temperature. Imaging was performed using an HC PL APO CS2 63X/ 1.4 NA objective at 5× digital zoom. The confocal pinhole was set at 1 Airy unit. Image scanning was done in bidirectional mode at 500 Hz. The excitation and emission filters for the mCherry channel were set to 552 nm and 576–651 nm, respectively. A circular region of radius 1.2 μm (bleached spot) at the cell edge was bleached using a 40 mW solid-state laser (100% intensity) for 500 ms, with a pixel dwell time of 1.15 μs. The laser powers for pre-bleach and post-bleach imaging were set at 1% (0.4 mW). 10 pre-bleached and 120 post-bleached frames were recorded in 512 × 512 pixel format at 2frame/s.

**FRAP analysis.** The measured raw fluorescence intensity was corrected for background fluorescence and photofading effects. At first, the fluorescence intensity at each frame was measured for the ROI defined by the bleached spot [$F_{ROI}(t)$]. To account for background fluorescence [$F_{bg}(t)$], the fluorescence was measured in a cell-free region of the image in the respective frame. To adjust for observational photofading [$F_{total}(t)$], the fluorescence from the total cell, except the bleached spot, was measured. The raw FRAP data [$F_{ROI}(t)$], were corrected for background and photofading at each frame by:

$$F_{corrected}(t) = \frac{F_{ROI}(t) - F_{bg}(t)}{F_{Total}(t) - F_{bg}(t)} \qquad (1)$$

Finally, the corrected FRAP ($F_{corrected}(t)$) was normalized by corrected prebleach intensity ($F'_{corrected}$):

$$F(t) = \frac{F_{corrected}(t)}{F'_{corrected}} \qquad (2)$$

$$\left[ F'_{corrected} = \frac{F_{ROI}(t_0) - F_{bg}(t_0)}{F_{Total}(t_0) - F_{bg}(t_0)} \right]$$

(Where $F_{ROI}(t_0)$, $F_{Total}(t_0)$, and $F_{bg}(t_0)$ indicate the initial fluorescence intensity from the bleached spot, total cell, and background, respectively)

The effective bleached spot radius ($r_e$) was determined (Supplementary Fig. 4h)[70]. The half-life of fluorescence recovery ($t_{1/2}$) was determined by fitting corrected fluorescence intensities at indicated time points to the first-order kinetic equation:

$$f(t) = A\left(1 - e^{-kt}\right) \left\{ where \; k = \frac{0.69}{t_{1/2}} \right\} \qquad (3)$$

Finally, the diffusion coefficient ($D_{confocal}$) was derived from the modified Soumpasis equation[106]:

$$D_{confocal} = 0.25 \frac{r_e^2}{t_{1/2}} \qquad (4)$$

where $r_e$ is the effective bleached spot radius and the coefficient 0.25 was numerically determined[70]

We validated our FRAP experimental setup by comparing the diffusion coefficients ($D_{confocal}$) of ligand-free EGFR-mCherry and VEGFR2-mCherry with previously reported diffusion coefficient measurements using single-molecule tracking. We observed that the diffusion coefficient of EGFR ($0.033 \, \mu m^2 \, s^{-1}$) and VEGFR2 ($0.020 \, \mu m^2 \, s^{-1}$)

(Table S3) determined from Confocal FRAP is in agreement with the average diffusion coefficient value reported for the heterogeneous population of EGFR ($0.036 \, \mu m^2 \, s^{-1}$)[96] and VEGFR2 ($0.032 \, \mu m^2 \, s^{-1}$)[107], respectively.

## Homology modelling

The models for the autoinhibited conformation of VEGFR1 (781-1158), VEGFR2 (787-1162), VEGFR1-JM$^{VEGFR2}$ chimera, and VEGFR1 triple mutated (K1079Q, K1142S, and R1146D) were built using iTasser server (https://zhanggroup.org/I-TASSER/)[108]. In all our models, the kinase domain adopts an autoinhibited conformation of C-helix-in and DFG−out. The kinase domain of VEGFR-1 (825-1158) was modelled based on PDB ID 3HNG, and the juxtamembrane segment (781-824) was modelled based on PDGFR structure (PDB ID 5K5X) (Supplementary Fig. 6a). The linker region between the N-lobe and C-lobe (925-991) could not be modelled due to the lack of any homology structure and remained unstructured in the model. The final model, with the highest confidence score as well as a low root-mean-square-deviation (see Table S5, RMSD of 0.589 Å w.r.t 3HNG) with respect to VEGFR1-structure (PDB ID: 3HNG), was selected and the inhibitor N-(4-chlorophenyl)−2-[(pyridin-4-ylmethyl)amino]benzamide was docked at the ATP binding pocket based on 3HNG structure, using PyMOL (DeLano, W. L., 2009). The structure was further energy minimized and equilibrated before analyzing by MD simulation. A preliminary MD simulation of 100 ns was performed to determine the structural stability of the selected model. The lowest energy structure selected remained stable in the preliminary simulation and was thus used for further studies. The models for the VEGFR1-JM$^{VEGFR2}$ chimera and the triple mutant construct of VEGFR1 were similarly constructed, energy minimized, and evaluated for further structural studies. Similarly, VEGFR2 (PDB ID 4AGC) was used to model the kinase domain (832-1162), and the juxtamembrane segment (787-831) was modelled based on PDGFR structure (PDB ID 5K5X). The model was selected as explained previously, and Axitinib was docked at the ATP binding pocket based on 4AGC structure, using PyMOL.

## Molecular dynamics simulations

The MD simulations were run using GROMACS 2019.6[109,110]. System preparation was done using the CHARMM-GUI server (www.charmm-gui.org/) with TIP3p water molecules[111] and 0.15 M NaCl. The model required 1000 steps approximately for energy minimization. The solvated system was equilibrated for 125 ps at a 1 fs time step, using H-bonds as constraints by implementing Linear Constraint Solver for Molecular Simulations (LINCS) algorithm[112]. At least three independent simulations of 2 μs each for VEGFR1 (wildtype), 500 ns each for VEGFR2 (wildtype), VEGFR1-JM$^{VEGFR2}$ chimera, and VEGFR1 triple mutated were performed using the CHARMM36m force field[113]. To comparatively study the orientation of the juxtamembrane segment with respect to the kinase domain in the crystal structure of cFMS (PDB ID 2OGV)[85] and FLT3 (PDB ID 1RJB)[20], we performed three independent simulation of 200 ns each. The simulations were performed under constant pressure (1 bar)[114] and temperature (300 K) (NPT)[115] and a time step of 2 fs. Potential-shift-Verlet was used for electrostatic and van der Waals interactions with a 12-Å cutoff. The trajectories were visually analyzed using VMD[116], and the structures were visualized in PyMol. RMSDs, RMSFs, and distances were measured using the tools provided in GROMACS.

## Statistics and reproducibility

Statistical analyses were performed with GraphPad Prism 9 using an unpaired two-tailed student's $t$-test. No statistical method was used to predetermine the sample size. No data were excluded from the analyses. All data were presented as mean ± SD from at least three biologically independent experiments. $P < 0.05$ was considered as statistically significant.

**Reporting summary**

Further information on research design is available in the Nature Portfolio Reporting Summary linked to this article.

## Data availability

The kinase domain structures used for the analysis of the conserved signature motifs in VEGFR1 and VEGFR2 can be found in the PDB: 3VHK (Crystal structure of the VEGFR2 kinase domain in complex with a back pocket binder), 1VR2 (Human vascular endothelial growth factor receptor 2 (KDR) Kinase domain) and 3HNG (Crystal structure of VEGFR1 in complex with N-(4-Chlorophenyl)−2-(pyridin-4-ylmethyl) amino)benzamide). The structure used for comparative structural analysis of VEGFR1 and VEGFR2 extracellular domain can be found in the PDB: 5T89 (Structure of VEGF-A in complex with VEGFR-1 domains D1-6). The JM segment of the VGEFR2 kinase domain (PDB: 4ASD (Crystal Structure of VEGFR2 Juxtamembrane and Kinase Domains) in Complex with SORAFENIB (BAY 43-9006) was modeled based on PDGFR crystal structure (PDB: 5K5X (Crystal structure of human PDGFRA)). The JM boundary of VEGFR2 was defined using FMS-like tyrosine kinase 3 structure (PDB: 1RJB (Crystal Structure of FLT3)). The kinase domain structures used to analyze the inhibitory interaction between the JM segment and activation loop in cFMS and VEGFR2 can be found in the PDB: 2OGV (Crystal Structure of the Autoinhibited Human c-Fms Kinase Domain) and 4AGC (Crystal structure of VEGFR2 juxtamembrane and kinase domains in complex with Axitinib (AG-013736) (N-Methyl-2-(3-((E)−2-pyridin-2-yl- vinyl)−1H-indazol-6-ylsulfanyl)-benzamide). All the relevant data are contained within this article and in the supporting information. Source data are available in the Source Data file and as a Figshare deposition (https://doi.org/10.6084/m9.figshare.24241207). Uncropped blots are available in the Source Data file. Source data are provided with this paper.

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

## Acknowledgements

The authors thank Dr. Arnab Gupta and his group members for access to the microscope facility. The authors thank Kaustav Gangopadhyay, Swarnendu Roy, and Subhankar Chowdhury for their help in the initial stage of the project and comments. The authors thank research funding from IISER Kolkata, infrastructural facilities supported by IISER Kolkata, and DST-FIST (SR/FST/LS-II/2017/93(c)). This work is supported by a

grant from SERB (CRG/2020/000437). Fellowships from CSIR-UGC supported MPC and PM; Fellowship from KVPY supported DD.

## Author contributions

The manuscript was written through the contribution of all authors. All authors have approved the final version of the manuscript. R.D. and M.P.C. designed the experiments. M.P.C., P.M., P.K., S.B., and P.K.D. performed the Biochemical experiments, Imaging, and data analysis. D.D. did the molecular dynamics simulation and data analysis. R.D. and M.P.C. wrote the manuscript.

## Competing interests

The authors declare no competing interests.

## Additional information

**Supplementary information** The online version contains Supplementary Material available at https://doi.org/10.1038/s41467-024-45499-2.

