## [Peer Review File · Nature Communications]

Molecular basis of VEGFR1 autoinhibition at the plasma membraneReviewer #1 (Remarks to the Author):

The submitted study is aimed at understanding aspects of the structure and function of the vascular endothelial growth factor receptor isoforms 1 and 2 (VEGFR1 and 2), key players in vasculogenesis/angiogenesis, and important drug targets. Data are presented that aim to resolve an outstanding question in the field, specifically why quite closely related VEGF receptor isoforms 1 and 2 (VEGFR1 and VEGFR2) have such different properties. In particular, the R1 form has very high affinity for growth factor but low kinase activity, whereas R2 has low VEGF affinity but high kinase activity. Despite these differences, both play critical roles in blood vessel growth and maintenance. As such, the work presented in the manuscript has the potential to be high impact since it touches on critical aspects of the activation mechanisms of the two receptor isoforms. More specifically, the study tries to explain why VEGFR1, despite being functionally critical, has weak kinase activity compared to its close relative VEGFR2. Like many RTKs, VEGFR2 undergoes ligand-independent activation at (unnaturally) high receptor density in the membrane, whereas VEGFR1 does not. Activation at high receptor density is presumably a stochastic process driven by increasing collisional frequency between monomers in the membrane. Why the two receptor isoforms should be different in this regard remains uncertain, and is addressed in this study. The methodology is fundamentally sound, using a similar approach to that used successfully to look at the mechanism of EGFR by the Kuriyan group (for example, in Endres et al, 2013). Variations in transient expression of different R1 and R2 receptor constructs are correlated with phosphorylation of specific tyrosine residues (detected by immunofluorescence) that in turn reports on receptor kinase activity. Critically, this assay depends on the quality (and specificity) of the antibodies used (see point raised, below).

As I see it, the key finding(s) of the study can be summarised:

1. It is known that the 'juxtamembrane region' (JM) controls kinase activity in a range of RTKs, acting as an 'autoinhibitory switch'. In this study, swapping the JM of VEGFR1 for that of VEGFR2 gives the R1 isoform a 'VEGFR2-like' property, high levels of growth factor-independent kinase activity at (unnaturally) high cell surface receptor densities.
2. The authors propose that the structure of the VEGFR1 JM is strongly autoinhibitory, making salt bridges with VEGFR1 kinase domain residues. Disruption of this interaction (via 'domain swapping' of the JM or kinase domain mutations) prevents autoinhibition, allowing kinase activation and ligand-independent autophosphorylation.
4. The authors conclude that the low kinase activity of VEGFR1 compared to VEGFR2 occurs as a consequence of a stronger interaction between the JM of VEGFR1 and its kinase domain.

In essence, the key finding of this study is interesting and would represent a valuable contribution. Unfortunately, the huge mass of data in the paper compromises the clarity and focus of the work. Some of the data presented either lack statistical significance or are not directly relevant to the main findings. My comments below are essentially aimed at removing a lot of the noise and refocusing the manuscript on the key findings, which do have significance.

Specific comments:

1. Figure 2. It would be useful to know what 'Low expressing' and 'High expressing' mean quantitatively – where do these sit on the x-axis of Fig. 2E, for example? In the microscopy images in A, C and D I note that there is no signal at all for phospho-tyrosine-specific antibodies, but none of the corresponding graphs showing a correlation between receptor levels and phosphorylation reach a zero value, hence there is an inconsistency between the microscopy images and the corresponding quantitation. Western blots with pY antibodies also consistently show signal at time zero. The authors should address this, since it somewhat undermines confidence in the assay.
2. It could be made clearer how the normalisation in Fig. 2E and F right panels has been performed.
3. Fig. 3E. It appears that only the dimerization control has produced any statistically significant effect, so this part of the figure is essentially redundant. I accept that there may be some kind of trend towards a ligand-dependent effect, but if it is not significant then it shouldn't be included. Same argument applies to Fig. 4F and 6E
4. Fig. 5. I don't find the MD simulations compelling – if the JM-S segment is intrinsically flexible and unresolved in xtal structures, isn't the MD simulation little more than a guess? At best, I see this as 'Supplemental' and a speculative discussion point.
5. Fig. 7. The authors moved on to include oxidative stress effects and a discussion of differential phosphatase effects – this seems unnecessary to me, causing the story to lose focus. The data

seem interesting, but including them (in my opinion) just causes the paper to lose focus and clarity.

6. There is far too much 'Supplementary' material, a lot of it just confirmatory and of little value. The manuscript would benefit from having much of this thinned out.

Reviewer #2 (Remarks to the Author):

SUMMARY

In this study, the authors aimed to understand the mechanisms of ligand-dependent and ligand-independent activation of VEGFR1. They investigated the differences between VEGFR1 and VEGFR2 in terms of auto-phosphorylation and stability, as well as the components responsible for the autoinhibition of VEGFR1. Additionally, they examined the conditions that induce autophosphorylation of VEGFR1 in pathological states. The findings of this study shed light on the regulatory mechanisms of VEGFR1 and have implications for the development of therapeutic strategies targeting VEGFRs.

The authors first compared the phosphorylation levels of VEGFR1 and VEGFR2 in the presence and absence of ligands. They demonstrated that VEGFR1 requires VEGF for phosphorylation, while VEGFR2 can be spontaneously phosphorylated even without ligand stimulation. Furthermore, they recapitulated prior work showing that the phosphorylation level of VEGFR1 after ligand stimulation was significantly lower than that of VEGFR2. To understand the underlying reasons for these differences, the authors investigated the phosphorylation kinetics and half-life of specific phosphotyrosine residues using immunoblotting. They discovered that the Y1213 residue in VEGFR1 is more transiently stable than the Y1179 residue in VEGFR2, indicating slower activation and a shorter half-life for VEGFR1. This is one major issue with the work, only one VEGFR1 site is examined—while there is a dearth of VEGFR1 antibodies, there are pan antibodies that could be used (non site specific). Therefore, their conclusions are only relevant to this one site, not to VEGFR1 as a whole.

To identify the component responsible for the autoinhibition of VEGFR1, the authors examined different regions of the receptor. They found that mutations or deletions in the extracellular domain (ECD) and transmembrane (TM) segment of VEGFR1 did not affect its autoinhibition. However, mutations in the juxtamembrane (JM) region of VEGFR1 partially or fully restored its autophosphorylation. This led the authors to conclude that the interactions between the TM and kinase domain are the main regulators of VEGFR1 autoinhibition.

Finally, the authors investigated the conditions that induce autophosphorylation of VEGFR1 in pathological states. They demonstrated that under H₂O₂ or sodium orthovanadate treatment, which mimic pathological conditions, wild-type VEGFR1 is autophosphorylated. This suggests that reactive oxygen species (ROS) and the inhibition of protein tyrosine phosphatase (PTP) play a role in VEGFR1 autophosphorylation.

Overall, this study provides valuable insights into the mechanisms underlying the autoinhibition and activation of VEGFR1. VEGFR1 is not widely studied, but it affects vascularization and even immune cell migration—so an improved understanding of its activation is sorely needed. The authors conducted a thorough investigation, using various experimental approaches, to elucidate these mechanisms. The findings have implications for the development of novel therapeutic strategies targeting VEGFRs. However, the title does seem to overstate the impact of the work, given the limited probing of VEGFR1 phosphorylation. Additionally, suggestions and questions regarding specific details and contextualization that could further enhance the manuscript are described below.

+++++

INTRODUCTION

• Page 1|line 7: In the first four sentences, the authors state that VEGFR1 is not autophosphorylated without ligand, unlike VEGFR2, and it shows weak tyrosine kinase activity even after the ligand stimulation. In the next sentence, they talk about VEGFR1 activation under pathological conditions. Since this sentence is after the "ligand stimulation" sentence, the phrase,

"which is puzzling", both comes across as awkward and is in conflict with the premise of the paper. It is well established that VEGFR1 transmits angiogenic signals (but to a lesser extent than VEGFR2), so referring to this concept as puzzling is not inline with conventional knowledge of VEGFR1's modulatory behavior. The manuscript could be improved if the concept of puzzling is removed.

- The paper mentions 'PDGFR-like JM-in inactive conformation' a few times. Two suggestions on this: (1) spell out PDGFR when introducing it for the first time (Page 3) and (2) provide background about this PDGFR JM-in inactive confirmation to contextualize its relevance in this VEGFR study.

- Page 1|line 10: I recommend adding the terminology "juxtamembrane" before mentioning JM.

- 'Downregulation' is more appropriate than 'deregulation' in this following context:

"Overexpression or deregulation of VEGFR1 is linked to several cancers and cancer-associated pain,...":

- The paper uses 'basal state' to refer to 'the absence of ligands' (Page 4). However, this usage could cause confusion because 'basal state' typically means 'a pre-activated or a resting condition'. In the absence of ligands, VEGFR2 can be activated, so the absence of ligands is not a basal state for VEGFR2. Additionally, this work examined the diseased condition (ROS) in the absence of ligands, which is not a basal state by conventional definition.

+++++

RESULTS

1. Page 5|line 1: The authors raise a very important concept that "receptor density at the plasma membrane is an important determinant of ligand-independent activation of RTKs." This is especially true for VEGFRs, where receptor densities of VEGFR1 and VEGFR2 have been sensitively quantified; however, the citations only point to either generalized RTK papers or papers on FGFR and EGFR, which are not relevant to this work. The manuscript could be improved by citing the relevant VEGFR papers that both cover this concept and quantify these receptor densities.

2. Page 5|lines 12-13: Rather than the terminology "peripheral region," terminology like perimembrane or cell perimeter could be used. Additionally, there should be more scientific insight into what specifically this would entail (e.g., cell membrane? How much distance into the cell). Furthermore, the peripheral (perimembrane) region may not be the right location for examining VEGFR1, since a majority of VEGFR1 is intracellular and a significant fraction localizes to the nucleus.

3. Page 5|lines 17-19: The manuscript reads "VEGFR2 linearly phosphorylates Y1175 upon ligand stimulation, suggesting the phosphorylation is independent of receptor concentration at the plasma membrane." The phosphorylation level of Y1175 increases as VEGFR2 concentration increases. More insight into why the phosphorylation is independent of receptor concentration would improve the paper.

4. Page 8|lines 7-10: In "The observed increase in the Dconfocal for the VEGFR1-GPA-G83I mutant ($0.033 \pm 0.013 \mu\text{m}^2\text{s}^{-1}$) confirms a dimer-to-monomer transition on mutating TM segment in the VEGFR1-GPA chimera ($0.021 \pm 0.005 \mu\text{m}^2\text{s}^{-1}$) (Figure 3E, S4G, and Table S3).", what is the criterion to determine if VEGFR tends to form a ligand-independent dimer?

5. Page 9|line 5: You mutated the ECD residue in VEGFR1 and saw that it didn't induce the VEGFR1's autophosphorylation. Could you explain more about how you get to the conclusion that the ECD is likely the dominant negative regulator of VEGFR1 activation?

6. Authors devised VEGFR1-only and VEGFR2-only transfected cells models to study VEGFR1 and VEGFR2 phosphorylation; however, these models do not consider the VEGFR1 and VEGFR2 phosphorylation through heterodimerization. This is an important mechanism enlisted by VEGFR1, so some context is required. Do the authors have comments on the heterodimerization mechanism? As a point of reference, it is well established that these receptors dimerize. Please see the two papers below, and there are others:

- o Autiero, M., Waltenberger, J., Communi, D. et al. Role of PIGF in the intra- and intermolecular cross talk between the VEGF receptors Flt1 and Flk1. *Nat Med* 9, 936–943 (2003).

<https://doi.org/10.1038/nm884>

- o Kui Huang, Charlotte Andersson, Godfried M. Roomans, Nobuyuki Ito, Lena Claesson-Welsh, Signaling properties of VEGF receptor-1 and -2 homo- and heterodimers, *The International Journal of Biochemistry & Cell Biology*, Volume 33, Issue 4, 2001, Pages 315-324, ISSN 1357-2725, [https://doi.org/10.1016/S1357-2725\(01\)00019-X](https://doi.org/10.1016/S1357-2725(01)00019-X).

7. This work focuses on only one phosphorylation site of each VEGFR: Y1213 for VEGFR1 and

Y1175 for VEGFR2. While we recognize that there are limited VEGFR1 antibodies available for each of the VEGFR1 sites, it is possible to look at pan-VEGFR1 phosphorylation. What is the authors' rationale for studying only these phosphorylation sites, not others or the total phosphorylation? Is it possible that other VEGFR1 sites can be auto-phosphorylated without the presence of VEGF?

8. Did the authors run a statistical analysis for the bar graphs in Figure 2 G and H? How many replicates (n) are included in these graphs?

9. Can the authors give more information about how to interpret the Dconfocal \rightarrow ? Is there a reference value for us to tell whether it's a dimer or monomer state? Is it a standardized measurement that can be compared across different studies?

10. Page 10: "The sequence comparison between VEGFR1 and VEGFR2 shows that the residues at the ligand-independent dimer interface are conserved. In contrast, the residues at the ligand-dependent dimer interface are not conserved." Have the authors considered whether this sequence difference could be responsible for the low ligand-dependent VEGFR1 phosphorylation stability?

11. Page 10: There seems to be a logical leap. What is the authors' rationale for the speculation that T763 and C764 make the TM segment incompatible with a ligand-independent dimer?

12. How are JM-B, JM-S, and JM-Z defined?

13. I am not sure if I understand how the conclusion was drawn "(VEGFR1) does not require help from a second tyrosine kinase".

14. Page 14 "We observed that wild-type VEGFR1 is autophosphorylated upon H2O2 treatment, but in the kinase-dead mutant, Y1213 was marginally phosphorylated (Figure 7A and S8A). Suggesting, under oxidative stress, VEGFR1 spontaneously autophosphorylates the Y1213 and does not require help from a second tyrosine kinase. In human colorectal cancer cells and hyperglycemia, it may be noted that the VEGFR1 and VEGFR2 phosphorylation is mediated by Src tyrosine kinase (80,84)."

+++++

CONCLUSION

1. This Conclusion section repeats the Results and did not speak much about the broader impact of these findings.
2. Can the authors provide more insights into the basis of this speculation? "We speculate that marginal reduction in PTP activity due to oxidative stress under pathological conditions may be sufficient to stimulate ligand-independent VEGFR1 signaling."

+++++

MINOR COMMENTS

1. Page 3, no period after (5,7,8)
2. Definition of graph (fig 2) should happen the first time the graph is used.
3. KD is conventionally used to describe equilibrium dissociation constant, this is not the appropriate use of this abbreviation. It suggested that a different abbreviation be used.
4. Page 6|line 22 (last line): Moving "The ligand bias by ..." to the next subsection might make the paragraphs more readable.
5. Page 7|lines 5-6: Could you match the illustration style showing VEGFR1 and VEGFR2 in Figure S3A so that the readers can compare their structures more easily?
6. Page 7|line 11: You might need to include a brief definition of "C482R".
7. Page 9|line 2: It is not clear if you mean "Figure S5A-B", or "Figure 5A-B".

Reviewer #3 (Remarks to the Author):

This study by Charaborty et al. provides a new finding, namely that the VEGFR1 juxtamembrane domain (JMD) has an autoinhibitory function that is not present in another related molecule,

VEGFR2. The study has much interest and relevance to VEGFR and receptor tyrosine kinase function. However, there are a number of issues which need to be carefully addressed. These points are listed below:

1) The authors indicate from the very first statement, that the low(er) levels of VEGFR1 tyrosine kinase (TK) activity is unusual; however, VEGFR2 is a notable and powerful TK. The lack of comparison to VEGFR3 (or any other RTK) in this context makes this statement shaky (compared to the 58 members of the human RTK family).

2) The overexpression of VEGFR-mCherry constructs raises serious mechanical issues in the context of RTK activation and possibly trafficking. Can the authors be confident that this is indeed similar to native or endogenous VEGFRs? The attachment of the 27 kDa mCherry to the C-terminal tail will restrict the movement of the flexible ~200 residue C-terminal tail that is likely to have regulatory effects on TK activation; furthermore VEGFR-mCherry trafficking may be modulated or disrupted. What controls have been done to check this?

3) The choice of phosphorylation epitopes in VEGFR1 (Y1213) and VEGFR2 (Y1175) is problematical. VEGFR2-Y1175 (vs. VEGFR1-Y1173) are sites of activation and binding to PLCgamma1 thus showing comparable properties; VEGFR2-Y1214 (vs. VEGFR1-Y1213) are linked to binding c-SRC upon phosphorylation and activation of downstream signalling pathways. Of note, VEGFR2-Y1175 and VEGFR2-Y1213 shows different kinetics upon activation by the same ligand, VEGF-A165 (Fearnley et al. (2016)). These experiments should be carried out by checking for total tyrosine phosphorylation using PY20 or PY-100 monoclonal antibodies. That will give a better idea of net activation resulting in the formation of total phosphotyrosine epitopes on the VEGFR1 vs VEGFR2 constructs.

4) The authors seem completely unaware of the multiple number of studies that indicate differential trafficking of VEGFR1 vs. VEGFR2. It is well established that VEGFR2 traffics slowly out of the secretory pathway, accumulating in the Golgi before reaching the plasma membrane and endosomes (Ewan et al., 2006; Manickam et al., 2011; Yamada et al., 2014). In contrast, conflicting reports suggest the majority of VEGFR1 localises to intracellular compartments such as Golgi (Mittar et al., 2009; Yang et al., 2015) and nucleus (Boulton et al., 2008; Zhang et al., 2010) with other reports of intracellular VEGFR1 near the nucleus (Lee et al., 2010). Weddell and Imoukhede (2017) have carried out mathematical modelling which predicts VEGFR1 distribution in endocytic vesicles, endosomes and nucleus but not necessarily at the plasma membrane. If resting VEGFR1 levels are largely (>80%) located within the cell, thus the fraction of VEGFR1 activation by exogenous ligand (e.g. VEGF-A) is relatively small. VEGFR1 is a widely expressed molecule, including epithelial cells with estimated numbers of 500-5000 molecules/cell. This is in contrast to endothelial VEGFR2 which is estimated at 10000-50000 molecules /cell. If VEGFR1 is retained (by virtue) of a targeting signal (e.g. within the JMD) to a different part of the cell, this may explain why VEGFR2 is much more readily available at the plasma membrane. This has not been considered.

5) The authors should remove Fig. 7 with experiments using hydrogen peroxide and phosphatase involvement. This does not add anything to the story and confuses the study as it stands.

6) The authors need to also carefully consider the work from the labs of Kuriyan, Lemmon and Schlessinger on studying EGFR and FGFR signalling and activation. There is much discussion on the roles of the flexible C-terminal tails and JMDs in influencing TK activation. This needs to be better placed within the introduction and discussion.

7) The whole manuscript, especially the figures, needs to be tidied up, data put into supplemental figures if needed, and a more tidy and streamlined article needs to be produced. Currently it feels that the authors have emptied their lab notebooks without any discretion. It does not make for easy reading or digestion by readers in the field.

Response to Reviewer #1:

We thank Reviewer#1 for evaluating our manuscript, and we are pleased to receive positive review about the quality of the work and the importance of the problem addressed in the current manuscript. We thankfully acknowledge the Reviewer for the complimentary notes. All the changes made in the text of the revised manuscript are highlighted with yellow.

Reviewer #1: Figure 2. It would be useful to know what 'Low expressing' and 'High expressing' mean quantitatively – where do these sit on the x-axis of Fig. 2E, for example? In the microscopy images in A, C and D I note that there is no signal at all for phospho-tyrosine-specific antibodies, but none of the corresponding graphs showing a correlation between receptor levels and phosphorylation reach a zero value, hence there is an inconsistency between the microscopy images and the corresponding quantitation. Western blots with pY antibodies also consistently show signal at time zero. The authors should address this, since it somewhat undermines confidence in the assay.

Author Response: We thank Reviewer #1 for critically reviewing the manuscript and thoughtful comments. Based on the suggestion we have now labelled the low and high-expressing cells in Figure 2E and 2F.

**Figure 2E and F:**

The expression level of VEGFR2 (panel E) or VEGFR1 (panel F) is plotted against the phosphorylation level of the corresponding tyrosine residues at the C-terminal tail. The low-expressing and high-expressing cells are indicated based on the mCherry intensity at the plasma membrane. Individual data points in the left panel represent the mean expression and phosphorylation level for the binned cells. The orange line represents the linear fitting of the individual data points in the ligand-dependent activation. The blue line in panel E represents the second-order polynomial fitting of the individual data points in the ligand-independent activation. In panel F, the blue line is the guiding line. The right panel represents the bar plot of the normalized phosphotyrosine levels. The phosphotyrosine level (FITC channel) is normalized with respect to the corresponding VEGFR expression level (mCherry channel) at the plasma membrane. The error bar shows the standard deviation of data points.

We thank the reviewer for pointing out this. To address the concern we determine the signal-to-noise ratio in the FITC (488 nm) channel of all the images in Figure 2A-D, because the phosphorylation level was determined from the intensity of the FITC channel. As shown in Figure R1 A, C, and D upper panel, the unstimulated cell has a background signal, which is approximately 10-fold less than the signal generated upon ligand stimulation. Since we plotted the raw FITC intensity in the Y-axis, the phosphorylation level does not reach zero (approx. background intensity is less than 2000 a.u.). However, activating the receptor by stimulating the cells with VEGF165 increases the FITC intensity by approximately 10 times.

Figure R1: Signal-to-noise analysis of Figure 2 A, B, C, D. The images in the middle panel showing the tyrosine phosphorylation (FITC 488 nm) is shown.

To understand the background signal found in the Western blot analysis, we transfected the CHO cell lines with increasing concentrations of VEGFR1 and VEGFR2 plasmid DNA (Figure R2). We next measure the phosphorylation of Y1213 or Y1175 with specific anti-phosphotyrosine antibodies. In the Western blot, VEGFR2 showed robust ligand-independent tyrosine phosphorylation when transfected with high DNA concentration. In contrast, VEGFR1 did not show any significant ligand-independent phosphorylation. In the kinetic studies, we used 750 μ g of DNA, the background signal we observe at zero time point for VEGFR2 most likely represents phosphotyrosine residues from high-expressing cells. This finding is consistent with the single-cell phosphorylation studies (Figure 2E and F).

Figure R2: Concentration dependent activation of VEGFR1 and VEGFR2. CHO cell was transfected with the increasing concentration of VEGFR1 and VEGFR2 plasmid to generate a low to high range of expression profile. In the upper panel, immunoblot shows the specific phosphorylation of indicated VEGFR constructs with and without ligand treatment. Bar plots in the lower panel represent the densitometric quantitation of the indicated phosphorylation level.

Reviewer #1: It could be made clearer how the normalisation in Fig. 2E and F right panels has been performed.

Author Response: We thank Reviewer#1 for the comments and observation. To address this point, we added the following lines to the figure caption of Figure 2 (E-F): ‘The right panel represents the bar plot of the normalized phosphotyrosine levels. The phosphotyrosine level (FITC channel) is normalized with respect to the corresponding VEGFR expression level (mCherry channel) at the plasma membrane’.

Reviewer # 1: Fig. 3E. It appears that only the dimerization control has produced any statistically significant effect, so this part of the figure is essentially redundant. I accept that there may be some kind of trend towards a ligand-dependent effect, but if it is not significant then it shouldn't be included. Same argument applies to Fig. 4F and 6E

Author Response: We thank the Reviewer for pointing out this. We have now added statistical analysis for all the dimer pairs in Figures 3E, 4F, and 6E.

Figure 3E: The diffusion coefficient measured from FRAP studies of indicated constructs of VEGFR1 and VEGFR2 in the presence and absence of VEGF₁₆₅ are plotted. VEGFR1-GPA chimera and VEGFR1-GPA-G83I chimera represent dimer and monomer controls, respectively. Each data point in the box plot reflects the diffusion coefficient of the selected cell, and the black line indicates the mean value. (For each construct, n=20-30 cells)

Figure 4F: The dimerization propensity of indicated VEGFR1 constructs is probed from the diffusion coefficient measured by the FRAP experiment. (For each construct, n=20-30 cells).

Figure 6E: The dimerization propensity of the indicated VEGFR1 construct is probed from the diffusion coefficient measured using the FRAP experiment. (For each construct, n=20-30 cells).

Reviewer #1: Fig. 5. I don't find the MD simulations compelling – if the JM-S segment is intrinsically flexible and unresolved in xtal structures, isn't the MD simulation little more than a guess? At best, I see this as 'Supplemental' and a speculative discussion point.

Author Response: We agree with Reviewer#1. We used the MD simulation for a speculative discussion point on the possible role of the JM interaction in stabilizing the autoinhibited conformation. Based on the suggestion, we have modified Figure 5 and moved the MD data to the Supporting Information.

Reviewer #1: Fig. 7. The authors moved on to include oxidative stress effects and a discussion of differential phosphatase effects – this seems unnecessary to me, causing the story to lose focus. The data seem interesting, but including them (in my opinion) just causes the paper to lose focus and clarity.

Author Response: Based on the suggestions we have removed Figure 7A-F, Figure S8 and the section "Cellular phosphatase balance modulates VEGFR-1 basal activation".

Reviewer #1: There is far too much 'Supplementary' material, a lot of it just confirmatory and of little value. The manuscript would benefit from having much of this thinned out.

Author Response: Thanks for pointing this out. We have now moved the data presented in the Supplementary material to the Source Data file provided along with this article. A detailed description is given below:

- Figure S1: Figures S1B-F are moved to the source data file. Instead, two new figures **S1B** and **S1C** are added.

- Figure S2: **Figure S2G** and bottom panel of Figure S2 G, D, and E are moved to the source data file.
- **Figure S3:** Main **Figure 3F-G** is moved to the supporting **Figure S3 F-G**.

- Figure S5: **Figure S5C-D** are moved to source data file. Statistical analysis added to new **panel D**.
- Figure S6: **Figure S6B-E and J** are removed and **Figure 5F-I** from main figure is added as supporting **Figure S6F-I**.
- Figure S7: 1. **Figure S6C-F** are moved to the source data file. The bottom panel of **Figure 6D** has been moved to supporting **Figure S7C**. The immunoblot from **Figure 6G** is moved to **Figure S7F**.
- Figure S8: Removed

-----Response to Reviewer # 1 End-----

Response to Reviewer #2:

We thank Reviewer #2 for the careful evaluation and for the helpful suggestion. All the changes made in the text are highlighted with yellow.

INTRODUCTION

Reviewer #2: Page 1|line 7: In the first four sentences, the authors state that VEGFR1 is not autophosphorylated without ligand, unlike VEGFR2, and it shows weak tyrosine kinase activity even after the ligand stimulation. In the next sentence, they talk about VEGFR1 activation under pathological conditions. Since this sentence is after the “ligand stimulation” sentence, the phrase, “which is puzzling”, both comes across as awkward and is in conflict with the premise of the paper. It is well established that VEGFR1 transmits angiogenic signals (but to a lesser extent than VEGFR2), so referring to this concept as puzzling is not inline with conventional knowledge of VEGFR1's modulatory behavior. The manuscript could be improved if the concept of puzzling is removed.

Author Response: We thank the Reviewer for the suggestion. We have now rephrased the sentence in the Abstract (line 5) and in the Introduction (Paragraph 3, line 1) section.

Reviewer #2: The paper mentions ‘PDGFR-like JM-in inactive conformation’ a few times. Two suggestions on this: (1) spell out PDGFR when introducing it for the first time (Page 3) and (2) provide background about this PDGFR JM-in inactive confirmation to contextualize its relevance in this VEGFR study.

Author Response: We thank the reviewer for pointing out this. We have now spelled out PDGFR in the second paragraph of the Introduction section and explained briefly the JM-in conformation. All the changes made are highlighted in yellow.

Reviewer #2: Page 1|line 10: I recommend adding the terminology “juxtamembrane” before mentioning JM.

Author Response: As suggested, we have now spelled out the terminology “juxtamembrane” in line 7 of the Abstract.

Reviewer #2: ‘Downregulation’ is more appropriate than ‘deregulation’ in this following context: “Overexpression or deregulation of VEGFR1 is linked to several cancers and cancer-associated pain,...”:

Author Response: Agreeing with the Reviewer we have replaced the ‘deregulation’ with ‘downregulation’ in the Introduction Paragraph 3 line 4.

Reviewer #2: The paper uses ‘basal state’ to refer to ‘the absence of ligands’ (Page 4). However, this usage could cause confusion because ‘basal state’ typically means ‘a pre-activated or a resting condition’. In the absence of ligands, VEGFR2 can be activated, so the absence of ligands is not a basal state for VEGFR2. Additionally, this work examined the diseased condition (ROS) in the absence of ligands, which is not a basal state by conventional definition.

Author Response:: Agreeing with the reviewer we have now referred to basal states as unligated or ligand-free states.

+++++

RESULTS

Reviewer #2: Page 5|line 1: The authors raise a very important concept that “receptor density at the plasma membrane is an important determinant of ligand-independent activation of RTKs.” This is especially true for VEGFRs, where receptor densities of VEGFR1 and VEGFR2 have been sensitively quantified; however, the citations only point to either generalized RTK papers or papers on FGFR and EGFR, which are not relevant to this work. The manuscript could be improved by citing the relevant VEGFR papers that both cover this concept and quantify these receptor densities.

Author Response:: We have now cited two papers mentioned below in the Result section, page no. 5:

56. Imoukhuede, P. I. & Popel, A. S. Quantification and cell-to-cell variation of vascular endothelial growth factor receptors. *Exp Cell Res* **317**, 955-965, doi:10.1016/j.yexcr.2010.12.014 (2011).

57. Imoukhuede, P. I. & Popel, A. S. Expression of VEGF receptors on endothelial cells in mouse skeletal muscle. *PLoS One* **7**, e44791, doi:10.1371/journal.pone.0044791 (2012).

Reviewer #2: Page 5|lines 12-13: Rather than the terminology “peripheral region,” terminology like perimembrane or cell perimeter could be used. Additionally, there should be more scientific insight into what specifically this would entail (e.g., cell membrane? How much distance into the cell). Furthermore, the peripheral (perimembrane) region may not be the right location for examining VEGFR1, since a majority of VEGFR1 is intracellular and a significant fraction localizes to the nucleus.

Author Response: We thank the reviewer for the suggestions. As explained in the Figure S1D of this manuscript, we now rephrase the sentence on page 5 paragraph 2, as ‘in this study, we focused on the regions surrounding the plasma membrane (cell perimeter).’ Figure R3 shows the distance (marked with dotted lines) into the line plot.

Figure R3: Selection of cell perimeter region for image analysis: (A) Confocal image of VEGFR1-mCherry expressed CHO cell line. The membrane (cell perimeter region) was stained with Wheat Germ Agglutinin(WGA) fused to Alexa 633. **(B)** Line plot shows the

selected region of the cell. **(C)** The distribution plot shows the average Width of the selected region for image analysis.

Previous studies suggest that in resting cells, a significant number of VEGFR1 (>80%) located within the cell, and a small fraction of VEGFR1 present on the plasma membrane could be activated by the exogenous ligand (Imoukhuede et al. *Exp Cell Res*, 2011). Our study is mainly focused on a general mechanism to understand how VEGFR1 remains autoinhibited in the cells. The JM autoinhibition presented in this study, in general, explains how the VEGFR1 remains in the off state at the plasma membrane and on the vesicular membrane inside the cell. The ligand-dependent activation of VEGFR1 and VEGFR2 could only be studied on the receptors localized to the plasma membrane. We considered that the conformation that stabilizes the active or inactive states of the kinase domain would be the same for the receptors located on the plasma membrane and the vesicular membrane inside the cell (Sarabipour et al., *eLife*, 2016). The immunoblot data using whole-cell lysate for studying phosphorylation may be limited by the cellular localization of the receptors. To find out if a common mechanism explains the autoinhibition of VEGFR1 inside the cell, we transfected the CHO cell lines with increasing concentrations of VEGFR1 and VEGFR2 plasmid DNA (Figure R4) and measured the phosphorylation of Y1213 or Y1175. In the Western blot, VEGFR2 showed robust ligand-independent tyrosine phosphorylation when transfected with high DNA concentration. In contrast, VEGFR1 did not show any significant ligand-independent phosphorylation. Also, the mutation studies (constitutive active mutants) (Figure 3C-D) and experiments with the chimeras (Figure 6C-D) using single-cell assay could be reproduced in the immunoblot experiment (Figure S3 D-G and Figure 4 B). Thus, our data suggests that the proposed mechanism of VEGFR1 autoinhibition universally explains how the kinase domain remains in an off state at the plasma membrane and inside the cell.

Figure R4: Concentration-dependent activation of VEGFR1 and VEGFR2. CHO cell was transfected with the increasing concentration of VEGFR1 and VEGFR2 plasmid to generate a low to high range of expression profile. In the upper panel, immunoblot shows the specific phosphorylation of indicated VEGFR constructs with and without ligand treatment. Bar plots in the lower panel represent the densitometric quantitation of the indicated phosphorylation level.

Reviewer #2: Page 5|lines 17-19: The manuscript reads “VEGFR2 linearly phosphorylates Y1175 upon ligand stimulation, suggesting the phosphorylation is independent of receptor concentration at the plasma membrane.” The phosphorylation level of Y1175 increases as VEGFR2 concentration increases. More insight into why the phosphorylation is independent of receptor concentration would improve the paper.

Author Response: Our data in Figure 2 E shows that in the absence of a ligand, VEGFR2 spontaneously autophosphorylates Y1175 at a critical concentration of the receptor at the membrane. Suggesting that the ligand-independent phosphorylation of Y1175 has a concentration dependency. However, in the presence of the ligand, we observed a linear increase in the phosphorylation of Y1175 as the concentration of the receptor increased on the membrane. There is no concentration dependency i.e., in the presence of a ligand the phosphorylation level is proportional to the receptor concentration at the plasma membrane. A similar observation was previously reported for EGFR by Endres, N. F. *et al. Cell* **152**, 543-556, doi:10.1016/j.cell.2012.12.032 (2013). We have now briefly explained this on page no.5 second paragraph.

Reviewer #2: Page 8|lines 7-10: In “The observed increase in the $D_{confocal}$ for the VEGFR1-GPA-G83I mutant ($0.033 \pm 0.013 \mu\text{m}^2\text{s}^{-1}$) confirms a dimer-to-monomer transition on mutating TM segment in the VEGFR1-GPA chimera ($0.021 \pm 0.005 \mu\text{m}^2\text{s}^{-1}$) (Figure 3E, S4G, and Table S3).”, what is the criterion to determine if VEGFR tends to form a ligand-independent dimer?

Author Response: In this study, we measure the relative changes in the diffusion coefficient ($D_{confocal}$) measured from the FRAP experiment to determine the oligomeric status of the receptor. We considered that a monomer would diffuse faster than a dimer (Chung *et al. Nature* 2010). For a dimer control, we turned to a glycoporphin-A (GPA), a canonical dimerization motif for the single-pass transmembrane domain (TM). The GPA was extensively studied by Engelman lab (Lemmon *et al. J Biol Chem.* 1992;267:7683–89, *Nature Struct Biol.* 1994;1:157–63. MacKenzie *et al. Science.* 1997;276:131–33 and Russ WP, Engelman DM. *PNAS.* 1999;96:863–68.) and is used as a model dimerization motif to study membrane protein dimer in a previous study by Hristova lab (Chen *et al J. Am. Chem. Soc.*, 132 (2010), pp. 3628-3635). For monomer control, we mutated Glycine at 83 position to Isoleucine (G83I) in the TM sequence of GpA, as suggested by (Sulistijo *et al. J Biol Chem*, 2003, 278: 51950-51956). The GpA dimer is mediated by the homotypic interaction of the G83 backbone. Mutating Glycine to isoleucine weakens the dimer interaction. In our studies, we observed that the monomer control VEGFR1-GPA-G83I mutant ($0.033 \pm 0.013 \mu\text{m}^2\text{s}^{-1}$) diffuses faster than the dimer control VEGFR1-GPA chimera ($0.021 \pm 0.005 \mu\text{m}^2\text{s}^{-1}$). Therefore, in our studies we compare all our data with respect to the monomer and dimer control.

Reviewer #2: Page 9|line 5: You mutated the ECD residue in VEGFR1 and saw that it didn't induce the VEGFR1's autophosphorylation. Could you explain more about how you get to the conclusion that the ECD is likely the dominant negative regulator of VEGFR1 activation?

Author Response: The extracellular domain inhibits the activation of the kinase domain in RTK. Studies from the Kuriyan, Lemmon, Schlessinger, and Hristova group have demonstrated that the ECD of EGFR, PDGFR, and FGFR play a negative role in receptor activation. Independent studies from Kuriyan (*Cell* 2013) and Leahy group (*Biochem J* 2020)

further showed that removing the ECD inhibition (by deletion) constitutively activates the kinase domain, which may be a conserved phenomenon in all seven classes of RTK. Together, studies from these groups suggest that the ECD is most likely the dominant negative regulator that prevents the spontaneous activation of RTK. We have now briefly explained this in the Result section. We have cited the following articles as references: 24, 65, 67, 73, 74, 75 and 76.

24. Markovic-Mueller, S. *et al.* Structure of the Full-length VEGFR-1 Extracellular Domain in Complex with VEGF-A. *Structure* **25**, 341-352, doi:10.1016/j.str.2016.12.012 (2017).
65. Tao, Q., Backer, M. V., Backer, J. M. & Terman, B. I. Kinase insert domain receptor (KDR) extracellular immunoglobulin-like domains 4-7 contain structural features that block receptor dimerization and vascular endothelial growth factor-induced signaling. *J Biol Chem* **276**, 21916-21923, doi:10.1074/jbc.M100763200 (2001).
67. Yuzawa, S. *et al.* Structural basis for activation of the receptor tyrosine kinase KIT by stem cell factor. *Cell* **130**, 323-334, doi:10.1016/j.cell.2007.05.055 (2007).
73. Arevalo, J. C. *et al.* A novel mutation within the extracellular domain of TrkA causes constitutive receptor activation. *Oncogene* **20**, 1229-1234, doi:10.1038/sj.onc.1204215 (2001).
74. Qiu, F. H. *et al.* Primary structure of c-kit: relationship with the CSF-1/PDGF receptor kinase family--oncogenic activation of v-kit involves deletion of extracellular domain and C terminus. *EMBO J* **7**, 1003-1011, doi:10.1002/j.1460-2075.1988.tb02907.x (1988).
75. Uren, A., Yu, J. C., Karcaaltincaba, M., Pierce, J. H. & Heidaran, M. A. Oncogenic activation of the alphaPDGFR defines a domain that negatively regulates receptor dimerization. *Oncogene* **14**, 157-162, doi:10.1038/sj.onc.1200810 (1997).
76. Gonzalez-Magaldi, M., McCabe, J. M., Cartwright, H. N., Sun, N. & Leahy, D. J. Conserved roles for receptor tyrosine kinase extracellular regions in regulating receptor and pathway activity. *Biochem J* **477**, 4207-4220, doi:10.1042/BCJ20200702 (2020).

Reviewer #2: Authors devised VEGFR1-only and VEGFR2-only transfected cells models to study VEGFR1 and VEGFR2 phosphorylation; however, these models do not consider the VEGFR1 and VEGFR2 phosphorylation through heterodimerization. This is an important mechanism enlisted by VEGFR1, so some context is required. Do the authors have comments on the heterodimerization mechanism? As a point of reference, it is well established that these receptors dimerize. Please see the two papers below, and there are others:

o Autiero, M., Waltenberger, J., Communi, D. *et al.* Role of PlGF in the intra- and intermolecular cross talk between the VEGF receptors Flt1 and Flk1. *Nat Med* **9**, 936-943 (2003). <https://doi.org/10.1038/nm884>

o Kui Huang, Charlotte Andersson, Godfried M. Roomans, Nobuyuki Ito, Lena Claesson-Welsh, Signaling properties of VEGF receptor-1 and -2 homo- and heterodimers, *The International Journal of Biochemistry & Cell Biology*, Volume 33, Issue 4, 2001, Pages 315-324, ISSN 1357-2725, [https://doi.org/10.1016/S1357-2725\(01\)00019-X](https://doi.org/10.1016/S1357-2725(01)00019-X).

Author Response: We thank the Reviewer for pointing to an important aspect of VEGFR1 activation. In the Discussion section of the current manuscript, we have cited the suggested articles and discussed this in view of biased signaling.

Reviewer #2: This work focuses on only one phosphorylation site of each VEGFR: Y1213 for VEGFR1 and Y1175 for VEGFR2. While we recognize that there are limited VEGFR1 antibodies available for each of the VEGFR1 sites, it is possible to look at pan-VEGFR1 phosphorylation. What is the authors' rationale for studying only these phosphorylation sites, not others or the total phosphorylation? Is it possible that other VEGFR1 sites can be auto-phosphorylated without the presence of VEGF?

Author Response: Thanks for the suggestions. As shown below in Figure R5, we have now repeated the phosphorylation kinetics of VEGFR1 and VEGFR2 after ligand stimulation and determined the total phosphotyrosine levels as a function of time. Our data suggests that the phosphorylation of VEGFR1 is transiently stable compared to sustained phosphorylation in VEGFR2. Overall, the total phosphorylation pattern follows the same trend as seen for the specific phosphotyrosine residue (Figure 2G). We observed that the VEGFR1 is transiently phosphorylated compared to the sustained phosphorylation of VEGFR2. Suggesting that the data for the phosphorylation of Y1213 or Y1175 agrees well with the total-phosphotyrosine antibodies. We have now included his results in Figures S1 B and S1C.

Figure R5 (Figure S1 B-C). Comparative analysis of VEGFR1 or VEGFR2 total phosphorylation level: (A-B) In the upper panel the immunoblot shows the total phosphorylation level of VEGFR1 (A) or VEGFR2 (B) at the indicated time points after activating the transfected CHO cell line with 50 nM VEGF₁₆₅. The lower panel shows the plot of the phosphorylation level of respective C-terminal tyrosine residue as a function of time. The phosphorylation level is analysed from the densitometric measurement of the Western blot shown in panel G. The $t_{1/2}$ is determined by fitting the decay of the highest intensity observed to exponential decay. The error bar shows the standard deviation from three independent experiments.

Reviewer #2: Did the authors run a statistical analysis for the bar graphs in Figure 2 G and H? How many replicates (n) are included in these graphs?

Author Response: Yes, we ran statistical analysis for the bar graphs in Figure 2 G and H. The data was replicated three times (n=3). All the statistical analyses are shown in Figure S2.

Reviewer #2: Can the authors give more information about how to interpret the D_{confocal} ? Is there a reference value for us to tell whether it's a dimer or monomer state? Is it a standardized measurement that can be compared across different studies?

Author Response: Various methods have been used by different laboratories to study the oligomerization of RTKs on the membrane. Some of the techniques are Fluorescence Recovery After Photobleaching (FRAP), Pulsed interleaved excitation fluorescence spectroscopy (PIE-FCCS), Single Molecule Tracking (SMT) and Förster resonance energy transfer (FRET). Among them, FRAP (Verveer et al., Science, 2000; Hillman and Schlessinger, Biochemistry, 1982; Schlessinger et al., PNAS, 1978; Kusumi et al., Biophysical Journal, 1993; Zidovetzki et al., PNAS, 1998), SMT (Freed et al, Cell, 2017; Rocha-Azevedo et al, Cell Rep, 2020; Sarah R. Needham et al., ncomms, 2016; Chung et al., Nature, 2010), and PIE-FCCS (Huang et al., elife, 2016; Shwetha Srinivasan et al., Nat. Comms, 2022) are used to determine the apparent diffusion coefficient of the receptor at the membrane. In all these studies it was considered that the monomeric state would diffuse faster relative to the dimer or oligomer.

In our study, we used a monomer and a dimer control of VEGFR1 to determine the relative change in diffusion coefficient (D_{confocal}). As anticipated, the dimer has a slower ($0.021 \pm 0.005 \mu\text{m}^2 \text{s}^{-1}$) diffusion rate compared to the faster diffusion of the monomer ($0.033 \pm 0.012 \mu\text{m}^2 \text{s}^{-1}$). Comparing the diffusion constant of unligated VEGFR2 ($0.020 \pm 0.007 \mu\text{m}^2 \text{s}^{-1}$) and the dimer control, we conclude that VEGFR2 remains as a preformed dimer even in the absence of a ligand (Figure 3E). This observation agrees with the previous independent studies using FRET (Sarabipour et al., Elife, 2016) and SMT (Bruno da Rocha-Azevedo et al., Cell Rep, 2020), which indicated that VEGFR2 forms a ligand-independent dimer on the membrane. The diffusion constant we measure for the VEGFR1 agrees with the overall observation from several independent studies with VEGFR2 and EGFR that dimer diffuses slower with respect to the monomer (fast) (see table below).

Technique	Receptor	Diffusion coefficient ($\mu\text{m}^2/\text{s}$)		Reference
		- ligand	+ ligand	
PIE-FCCS	EGFR	0.60 ± 0.02	0.36 ± 0.01	Huang et al., elife, 2016
SMT	EGFR	0.036	0.014	Freed et al, Cell, 2017
SMT	VEGFR2	0.032		Bruno da Rocha-Azevedo et al, Cell Rep, 2020
SMT	EGFR	0.01-0.02		Sarah R. Needham et al., ncomms, 2016
FRAP	VEGFR2	0.19	0.09	Valentina Damioli et al., Scientific Reports, 2017
SMT	EGFR	0.17 ± 0.06	0.08 ± 0.03	Chung et al., Nature, 2010

Reviewer #2: Page 10: “The sequence comparison between VEGFR1 and VEGFR2 shows that the residues at the ligand-independent dimer interface are conserved. In contrast, the residues at the ligand-dependent dimer interface are not conserved.” Have the authors considered whether this sequence difference could be responsible for the low ligand-dependent VEGFR1 phosphorylation stability?

Author Response: Yes we considered if the sequence difference in the TM segment is responsible for the low phosphorylation of VEGFR1. To test that we performed three

experiments: in one experiment we replaced (by point mutation) the residues at the active TM dimer interface of VEGFR1 by VEGFR2 one at a time (Figure S5C), secondly replaced the TM segment of VEGFR1 with canonical TM dimer segment of GPA (Figure S4 I), and finally replaced the TM segment of VEGFR1 with VEGFR2 (Figure 4B, C, and E). We observed that in none of the experiments, VEGFR1 was spontaneously phosphorylated. Together, our data suggest that the sequence difference in the TM segment could not explain the difference in the activation of VEGFR1 and VEGFR2. However, replacing the JM segment of VEGFR1 with VEGFR2 (Figure 4B-C) spontaneously activates VEGFR1 (Figure 6D).

Figure S5C : Top panel is the Immunoblot of Y1213 phosphorylation in transmembrane domain mutants and chimeric constructs of VEGFR1. The densitometric quantification of Y1213Y phosphorylation is depicted by bar graphs (bottom panel).

Figure S4I: Immunoblot (left panel) showing the Y1213 phosphorylation level of VEGFR1-TM^{GPA} (TM dimer control) as compared to its wild-type control. Bar plots in the right panel represent the quantitation of the Y1213 phosphorylation level.

Figure 4. Functional analysis of TM and JM segments in ligand-independent activation of VEGFR1

(B) Immunoblot showing the phosphorylation of Y1213 in the indicated constructs of VEGFR1. The expression level of the VEGFR1 is determined using an antiHA antibody. The bar plot in the lower panel represents the relative Y1213 phosphorylation level determined from densitometric analysis. The data represent the mean \pm SD (n=3).

(C) The plot of the Y1213 phosphorylation level against the expression level of VEGFR1 Δ ECD and wt from the single-cell assay.

(E) The plot of Y1213 phosphorylation versus the expression level of indicated VEGFR1 constructs.

Figure 6D: The plot of Y1213 phosphorylation versus the expression level of indicated VEGFR1 constructs.

Reviewer #2: Page 10: There seems to be a logical leap. What is the authors' rationale for the speculation that T763 and C764 make the TM segment incompatible with a ligand-independent

dimer?

Author Response: Previous studies from Engelman and colleagues, and Maruyama and colleagues showed that threonine and cystine could drive the association of TM helix dimer. We speculate that in the case of VEGFR1 threonine and cystine at the TM dimer interface would bias the structure more to the active conformation, rendering ligand-independent dimer less stable. In this respect, we have discussed and cited three papers in the result section as listed below.

81. Dawson, J. P., Weinger, J. S. & Engelman, D. M. Motifs of serine and threonine can drive association of transmembrane helices. *J Mol Biol* **316**, 799-805, doi:10.1006/jmbi.2001.5353 (2002).
82. Moriki, T., Maruyama, H. & Maruyama, I. N. Activation of preformed EGF receptor dimers by ligand-induced rotation of the transmembrane domain. *J Mol Biol* **311**, 1011-1026, doi:10.1006/jmbi.2001.4923 (2001).
83. Krimmer, S. G. *et al.* Cryo-EM analyses of KIT and oncogenic mutants reveal structural oncogenic plasticity and a target for therapeutic intervention. *Proc Natl Acad Sci U S A* **120**, e2300054120, doi:10.1073/pnas.2300054120 (2023).

Reviewer #2: How are JM-B, JM-S, and JM-Z defined?

Author Response: The JM-B, JM-S, and JM-Z are defined based on the FLT3 structure. We have cited the below article in Figure 5A.

20. Griffith, J. *et al.* The structural basis for autoinhibition of FLT3 by the juxtamembrane domain. *Mol Cell* **13**, 169-178, doi:10.1016/s1097-2765(03)00505-7 (2004).

Reviewer #2: I am not sure if I understand how the conclusion was drawn “(VEGFR1) does not require help from a second tyrosine kinase”.

Author Response: In Figure R6, we determined the phosphorylation of VEGFR1 kinase-dead mutant (D1022N) before and after ligand stimulation. Our data suggests that kinase activity of VEGFR1 is required for the autophosphorylation of the C-terminal tail and it does not depend on the other tyrosine kinases, like Src. If VEGFR1 is phosphorylated by Src, then we would expect phosphorylation for the kinase-dead mutant.

However, based on the suggestions made by Reviewer 1 and Reviewer 3, we have removed this section from the current revised manuscript.

Figure R6: Representative immunoblot of VEGFR1 1213 phosphorylation level compared to kinase-dead mutant(D1022N) at the indicated experimental condition. Below is the densitometric analysis of 112313 phosphorylation level for the indicated VEGFR1 constructs

Reviewer #2: Page 14: “We observed that wild-type VEGFR1 is autophosphorylated upon H₂O₂ treatment, but in the kinase-dead mutant, Y1213 was marginally phosphorylated (Figure 7A and S8A). Suggesting, under oxidative stress, VEGFR1 spontaneously autophosphorylates the Y1213 and does not require help from a second tyrosine kinase. In human colorectal cancer cells and hyperglycemia, it may be noted that the VEGFR1 and VEGFR2 phosphorylation is mediated by Src tyrosine kinase (80,84).”

Author Response: Based on the suggestions made by Reviewer 1 and Reviewer 2, we have removed this section from the current revised manuscript.

+++++

CONCLUSION

Reviewer #2:

1. This Conclusion section repeats the Results and did not speak much about the broader impact of these findings.

2. Can the authors provide more insights into the basis of this speculation? “We speculate that marginal reduction in PTP activity due to oxidative stress under pathological conditions may be sufficient to stimulate ligand-independent VEGFR1 signaling.”

Author Response: Based on the suggestion made by Reviewer#1, Reviewer#3, and Senior editor we have separately written the Discussion section.

Since cellular protein tyrosine phosphatases (PTP) remove the phosphate group from the phosphoserine residues, they are an important player in maintaining equilibrium. Thus, a subtle imbalance in phosphatase activity (inhibition) may perturb the equilibrium state. In this respect, we have discussed the following two papers:

89. Bae, Y. S. *et al.* Epidermal growth factor (EGF)-induced generation of hydrogen peroxide: role in EGF receptor-mediated tyrosine phosphorylation. *Journal of Biological Chemistry* **272**, 217-221 (1997).
90. Sundaresan, M., Yu, Z. X., Ferrans, V. J., Irani, K. & Finkel, T. Requirement for generation of H₂O₂ for platelet-derived growth factor signal transduction. *Science* **270**, 296-299, doi:10.1126/science.270.5234.296 (1995).

++++
MINOR COMMENTS

Reviewer #2: Page 3, no period after (5,7,8)

Author Response: We have corrected the reference

Reviewer #2: Definition of graph (fig 2) should happen the first time the graph is used.

Author Response: We have defined Figure 2 on page no. 5, paragraph 2 and the Figure caption.

Reviewer #2: KD is conventionally used to describe equilibrium dissociation constant, this is not the appropriate use of this abbreviation. It suggested that a different abbreviation be used.

Author Response: We have now spelled the kinase domain.

Reviewer #2: Page 6|line 22 (last line): Moving “The ligand bias by ...” to the next subsection might make the paragraphs more readable.

Author Response:: As suggested by the senior editor we have separated the Result and Discussion. The above-mentioned section is now part of the discussion.

Reviewer #2: Page 7|lines 5-6: Could you match the illustration style showing VEGFR1 and VEGFR2 in Figure S3A so that the readers can compare their structures more easily?

Author Response: The schematic diagram of VEGFR1 ECD is drawn based on the high-resolution full-length cryo-EM structure (PDB: ID 5T89). However, due to a lack of a high-resolution full-length structure of VEGFR2, the schematic is drawn based on a low-resolution negative stain electron micrograph. The following articles are cited as references:

18. Markovic-Mueller, S. *et al.* Structure of the Full-length VEGFR-1 Extracellular Domain in Complex with VEGF-A. *Structure* **25**, 341-352, doi:10.1016/j.str.2016.12.012 (2017).
19. Ruch, C., Skiniotis, G., Steinmetz, M. O., Walz, T. & Ballmer-Hofer, K. Structure of a VEGF-VEGF receptor complex determined by electron microscopy. *Nat Struct Mol Biol* **14**, 249-250, doi:10.1038/nsmb1202 (2007).

Reviewer #2: Page 7|line 11: You might need to include a brief definition of “C482R”.

Author Response: We have now briefly described the C482R mutation on page no. 7.

Reviewer #2: Page 9|line 2: It is not clear if you mean “Figure S5A-B”, or “Figure 5A-B”

Author Response: Thank you for pointing out the typo. We have now corrected the typo.

-----Response to Reviewer # 2 End-----

Response to Reviewer #3:

We thank Reviewer #3 for the careful evaluation and for the helpful suggestion. All the changes made in the text of the revised manuscript are highlighted with yellow.

Reviewer #3: The authors indicate from the very first statement, that the low(er) levels of VEGFR1 tyrosine kinase (TK) activity is unusual; however, VEGFR2 is a notable and powerful TK. The lack of comparison to VEGFR3 (or any other RTK) in this context makes this statement shaky (compared to the 58 members of the human RTK family).

Author Response: We thank the Reviewer for the suggestions. We have now modified the Abstract and the Introduction section accordingly.

Reviewer #3: The overexpression of VEGFR-mCherry constructs raises serious mechanical issues in the context of RTK activation and possibly trafficking. Can the authors be confident that this is indeed similar to native or endogenous VEGFRs? The attachment of the 27 kDa mCherry to the C-terminal tail will restrict the movement of the flexible ~200 residue C-terminal tail that is likely to have regulatory effects on TK activation; furthermore, VEGFR-mCherry trafficking may be modulated or disrupted. What controls have been done to check this?

Author Response: We thank the Reviewer for the thoughtful suggestions. Given the advantage of using mCherry fused to the C-terminal tail of RTK in live cells and in fixed cells, this technique is used for studying other RTKs like FGFR and EGFR (Endres, N. F. *et al.*, Cell, 2013; Chung, I. *et al.*, Nature, 2010; Sarabipour et al., Elife, 2016; Freed et al., Cell, 2017)

To evaluate the possibility of interference by the 27kDa mCherry attached to the C-terminal tail, we measure the: 1) localization of the VEGFR constructs (with and without mCherry (- Δ mCherry)) to the plasma membrane, 2) determined the ligand-dependent activity and 3) measured the phosphorylation kinetics (Figure R7). To find the plasma membrane localization of VEGFR, we labeled the membrane with Wheat Germ Agglutinin tagged with Alexa 633 and determined the Pearson correlation coefficient (PCC) with the VEGFR attached to mCherry or VEGFR- Δ mCherry localized to the plasma membrane- (Figure R7 A-B). The VEGFR- Δ mCherry was visualized by immunostaining with an anti-VEGFR antibody followed by a secondary antibody attached to FITC. The high PCC values for the VEGFR-mCherry (Figure R8) and VEGFR- Δ mCherry (Figure R7C) construct suggest that both forms of receptors are equally localized to the membrane. The immunoblotting in Figure R7D-E shows that both the VEGFR-mCherry and VEGFR- Δ mCherry are stimulated equally with VEGF₁₆₅. Finally, we measure the phosphorylation kinetics of the tyrosine residues in the C-terminal tail of VEGFR- Δ mCherry after ligand stimulation (Figure R7 F-G). We observed that tyrosine phosphorylation of the VEGFR- Δ mCherry construct follows the same trend as that of the VEGFR-mCherry construct in Figure 2G-H. Based on our data, we conclude that both the VEGFR construct (with and without mCherry) have similar plasma membrane localization and could be activated in the same fashion. Moreover, we would like to point out that a similar construct of VEGFR2-mCherry was used in previous studies (Sarabipour et al. *Elife* 2016).

Figure R7. Comparative analysis of VEGFR1 or VEGFR2 localization and activation in the presence and absence of mCherry tag

(A-B) Confocal imaging of mCherry deleted VEGFR1 (A) or VEGFR2 (B) expressed in CHO cell lines. The membrane was stained with Wheat Germ Agglutinin fused to Alexa 633. The protein was immunostained with respective anti-VEGFR antibody and visualized with a secondary antibody tagged with FITC. The first row is the Immunostaining of transfected VEGFR (left), WGA is in the middle row, and the corresponding green/magenta overlay is in the right. (Scale bar = 10 μ m).

(C) Quantification of colocalization between VEGFR2 (n= 15) or VEGFR1 (n= 16) and WGA by Pearson's correlation coefficient.

(D-E) In the upper panel, the Immunoblot shows the phosphorylation of Y1175(D) or Y1213(E) in the indicated constructs of VEGFR1 or VEGFR2 respectively. The bar plot in the lower panel represents the relative Y1175(D) or Y1213(E) phosphorylation level determined from densitometric analysis. The data represent the mean \pm SD (n=2).

(F-G) In upper panel The immunoblot shows the representative phosphorylation level of VEGFR1 or VEGFR2 at the indicated time points after activating the transfected CHO cell line with 50nM VEGF₁₆₅. Lower panel shows the plot of the phosphorylation level of respective C-terminal tyrosine residue as a function of time. The phosphorylation level is analyzed from the densitometric measurement of the Western blot shown in the upper panel.

Figure R8. Comparative analysis of VEGFR1 or VEGFR2 localization at plasma membrane

(A-B) Confocal imaging of mCherry fused VEGFR1 (A) or VEGFR2 (B) expressed in CHO cell lines. The membrane was stained with Wheat Germ Agglutinin fused to Alexa 633. The receptor at the membrane was visualized by red channel ($\lambda_{\text{ex}} = 552\text{nm}$, $\lambda_{\text{em}} = 586\text{-}651\text{nm}$). The first row is the Immunostaining of transfected VEGFR (left), WGA is in the middle row, and the corresponding green/magenta overlay is in the right. (Scale bar = 10 μm)

(C) Quantification of colocalization between VEGFR2 (n= 36) or VEGFR1 (n= 32) and WGA by Pearson's correlation coefficient..

- Endres, N. F. *et al.* Conformational coupling across the plasma membrane in activation of the EGF receptor. *Cell* **152**, 543-556, doi:10.1016/j.cell.2012.12.032 (2013).
- Chung, I. *et al.* Spatial control of EGF receptor activation by reversible dimerization on living cells. *Nature* **464**, 783-787, doi:10.1038/nature08827 (2010).
- Sarabipour, S., Ballmer-Hofer, K. & Hristova, K. VEGFR-2 conformational switch in response to ligand binding. *Elife* **5**, e13876, doi:10.7554/eLife.13876 (2016).
- Freed, D. M. *et al.* EGFR Ligands Differentially Stabilize Receptor Dimers to Specify Signaling Kinetics. *Cell* **171**, 683-695 e618, doi:10.1016/j.cell.2017.09.017 (2017).

Reviewer #3: The choice of phosphorylation epitopes in VEGFR1 (Y1213) and VEGFR2 (Y1175) is problematical. VEGFR2-Y1175 (vs. VEGFR1-Y1173) are sites of activation and binding to PLCgamma1 thus showing comparable properties; VEGFR2-Y1214 (vs. VEGFR1-Y1213) are linked to binding c-SRC upon phosphorylation and activation of downstream signalling pathways. Of note, VEGFR2-Y1175 and VEGFR2-Y1213 shows different kinetics upon activation by the same ligand, VEGF-A165 (Fearnley et al. (2016)). These experiments should be carried out by checking for total tyrosine phosphorylation using PY20 or PY-100 monoclonal antibodies. That will give a better idea of net activation resulting in the formation of total phosphotyrosine epitopes on the VEGFR1 vs VEGFR2 constructs.

Author Response:: We thank the reviewer for thoughtful comments. To address the point raised by the reviewer, we study the phosphorylation kinetics of VEGFR1 and VEGFR2 with total phosphotyrosine antibodies (Figure R9). In the experiment, we measured the phosphorylation level of the receptor after ligand stimulation for one hour. We observed that

the total phosphorylation follows the same trend as shown in Figure 2G-H. The VEGFR1 phosphorylate transiently, whereas VEGFR2 showed sustained phosphorylation. We have included this data in Figure S1 B-C of the manuscript and discussed in the result section.

Figure R9. Comparative analysis of VEGFR1 or VEGFR2 total phosphorylation level (A-B) In the upper panel, the immunoblot shows the total phosphorylation level of VEGFR1(A) or VEGFR2(B) at the indicated time points after activating the transfected CHO cell line with 50 nM VEGF₁₆₅. The lower panel shows the plot of the total phosphorylation level as a function of time. The phosphorylation level is analyzed from the densitometric measurement of the Western blot shown in the upper panel. The $t_{1/2}$ is determined by fitting the decay of the highest intensity observed to exponential decay. The error bar represents the standard deviation from three independent experiments.

Reviewer #3: The authors seem completely unaware of the multiple number of studies that indicate differential trafficking of VEGFR1 vs. VEGFR2. It is well established that VEGFR2 traffics slowly out of the secretory pathway, accumulating in the Golgi before reaching the plasma membrane and endosomes (Ewan et al., 2006; Manickam et al., 2011; Yamada et al., 2014). In contrast, conflicting reports suggest the majority of VEGFR1 localises to intracellular compartments such as Golgi (Mittar et al., 2009; Yang et al., 2015) and nucleus (Boulton et al., 2008; Zhang et al., 2010) with other reports of intracellular VEGFR1 near the nucleus (Lee et al., 2010). Weddell and Imoukhede (2017) have carried out mathematical modelling which predicts VEGFR1 distribution in endocytic vesicles, endosomes and nucleus but not necessarily at the plasma membrane. If resting VEGFR1 levels are largely (>80%) located within the cell, thus the fraction of VEGFR1 activation by exogenous ligand (e.g. VEGF-A) is relatively small. VEGFR1 is a widely expressed molecule, including epithelial cells with estimated numbers of 500-5000 molecules/cell. This is in contrast to endothelial VEGFR2 which is estimated at 10000-50000 molecules /cell. If VEGFR1 is retained (by virtue) of a targeting signal (e.g. within the JMD) to a different part of the cell, this may explain why VEGFR2 is much more readily available at the plasma membrane. This has not been considered.

Author Response: We thank the Reviewer for the suggestion. Indeed, the reviewer raised an important point, particularly regarding VEGFR localization and trafficking. In this study, we focused on the VEGFR1 and VEGFR2 that are localized to the plasma membrane. To find out if VEGFR1 and VEGFR2 are localizing to the same extent to the plasma membrane, we compare the Pearson correlation coefficient (PCC) of VEGFR1 and VEGFR2 with respect to the membrane marked with Wheat Germ Agglutinin tagged with Alexa 633. As shown in Figure R8, for cells expressing a comparable amount of receptors at the membrane, we observed that under our experimental condition, both VEGFR1 and VEGFR2 are equally localizing to the plasma membrane. Since we focused on the plasma membrane to evaluate the structure-function relation, the single-cell experiments thus are not biased by the differential localization of the VEGFR1 or VEGFR2. In the single-cell assay, we compared cells that express equal levels of VEGFR1 and VEGFR2 on the plasma membrane. Therefore, the differential phosphorylation of VEGFR1 and VEGFR2 reflects their difference in the catalytic activity of the kinase domain.

The proposed autoinhibitory mechanism of the VEGFR1 presents a general principle explaining how the JM segment stabilizes the inactive conformation of the kinase domain. The model developed on the VEGFR1 expressed on the plasma membrane could be extended to the VEGFR1 localized on the other vesicular membrane in the cell (Sarabipour et al., eLife,2016). The immunoblot data using the whole-cell lysate for studying phosphorylation may be limited by the cellular localization of the receptors. To test that, we transfected the CHO cell lines with increasing concentrations of VEGFR1 and VEGFR2 plasmid DNA (Figure R10) and measured the phosphorylation of Y1213 or Y1175. In the Western blot, VEGFR2 showed robust ligand-independent tyrosine phosphorylation when transfected with high DNA concentration. In contrast, VEGFR1 did not show any significant ligand-independent phosphorylation. Together, our data suggests that the proposed mechanism of VEGFR1 autoinhibition universally explains how the kinase domain remains in an off state at the plasma membrane and inside the cell.

Figure R10: Concentration-dependent activation of VEGFR1 and VEGFR2. CHO cell was transfected with the increasing concentration of VEGFR1 and VEGFR2 plasmid DNA to generate a low to high range of expression profile. In the upper panel, immunoblot shows the specific phosphorylation of indicated VEGFR constructs with and without ligand treatment. Bar plots in the lower panel represent the densitometric quantitation of the indicated phosphorylation level.

Reviewer #3: The authors should remove Fig. 7 with experiments using hydrogen peroxide and phosphatase involvement. This does not add anything to the story and confuses the study as it stands.

Author Response: Based on the suggestions we have removed Figure 7A-F, Figure S8, and the section “Cellular phosphatase balance modulates VEGFR-1 basal activation”.

Reviewer #3: The authors need to also carefully consider the work from the labs of Kuriyan, Lemmon and Schlessinger on studying EGFR and FGFR signalling and activation. There is much discussion on the roles of the flexible C-terminal tails and JMDs in influencing TK activation. This needs to be better placed within the introduction and discussion.

Author Response: Thank you for the suggestions. Now we have discussed following papers from the Kuriyan, Lemmon and Schlessinger groups in the Introduction, Results, and Discussion section.

1. Kuriyan Lab:

- Zhang, X., Gureasko, J., Shen, K., Cole, P. A. & Kuriyan, J. An allosteric mechanism for activation of the kinase domain of epidermal growth factor receptor. *Cell* **125**, 1137-1149, doi:10.1016/j.cell.2006.05.013 (2006).
- Kovacs, E. *et al.* Analysis of the Role of the C-Terminal Tail in the Regulation of the Epidermal Growth Factor Receptor. *Mol Cell Biol* **35**, 3083-3102, doi:10.1128/MCB.00248-15 (2015).
- Jura, N. *et al.* Mechanism for activation of the EGF receptor catalytic domain by the juxtamembrane segment. *Cell* **137**, 1293-1307, doi:10.1016/j.cell.2009.04.025 (2009).
- Jura, N. *et al.* Catalytic control in the EGF receptor and its connection to general kinase regulatory mechanisms. *Mol Cell* **42**, 9-22, doi:10.1016/j.molcel.2011.03.004 (2011).
- Endres, N. F. *et al.* Conformational coupling across the plasma membrane in activation of the EGF receptor. *Cell* **152**, 543-556, doi:10.1016/j.cell.2012.12.032 (2013).
- Huang, Y. *et al.* A molecular mechanism for the generation of ligand-dependent differential outputs by the epidermal growth factor receptor. *Elife* **10**, doi:10.7554/eLife.73218 (2021).
- Huang, Y. *et al.* Molecular basis for multimerization in the activation of the epidermal growth factor receptor. *Elife* **5**, doi:10.7554/eLife.14107 (2016).

2. Lemmon Lab

- Lemmon, M. A., Flanagan, J. M., Treutlein, H. R., Zhang, J. & Engelman, D. M. Sequence specificity in the dimerization of transmembrane. alpha-helices. *Biochemistry* **31**, 12719-12725 (1992).
- Lemmon, M. A. *et al.* Glycophorin A dimerization is driven by specific interactions between transmembrane alpha-helices. *Journal of Biological Chemistry* **267**, 7683-7689, doi:[https://doi.org/10.1016/S0021-9258\(18\)42569-0](https://doi.org/10.1016/S0021-9258(18)42569-0) (1992).
- Freed, D. M. *et al.* EGFR Ligands Differentially Stabilize Receptor Dimers to Specify Signaling Kinetics. *Cell* **171**, 683-695 e618, doi:10.1016/j.cell.2017.09.017 (2017).
- Lemmon, M. A., Freed, D. M., Schlessinger, J. & Kiyatkin, A. The Dark Side of Cell Signaling: Positive Roles for Negative Regulators. *Cell* **164**, 1172-1184, doi:10.1016/j.cell.2016.02.047 (2016).
- Kiyatkin, A., van Alderwerelt van Rosenburgh, I. K., Klein, D. E. & Lemmon, M. A. Kinetics of receptor tyrosine kinase activation define ERK signaling dynamics. *Sci Signal* **13**, doi:10.1126/scisignal.aaz5267 (2020).
- Lemmon, M. A. *et al.* Two EGF molecules contribute additively to stabilization of the EGFR dimer. *EMBO J* **16**, 281-294, doi:10.1093/emboj/16.2.281 (1997).
- Red Brewer, M. *et al.* The juxtamembrane region of the EGF receptor functions as an activation domain. *Mol Cell* **34**, 641-651, doi:10.1016/j.molcel.2009.04.034 (2009).

3. Schlessinger Lab

- Lemmon, M. A. & Schlessinger, J. Cell signaling by receptor tyrosine kinases. *Cell* **141**, 1117-1134, doi:10.1016/j.cell.2010.06.011 (2010).
- Yarden, Y. & Schlessinger, J. Epidermal growth factor induces rapid, reversible aggregation of the purified epidermal growth factor receptor. *Biochemistry* **26**, 1443-1451, doi:10.1021/bi00379a035 (1987).
- Bae, J. H. & Schlessinger, J. Asymmetric tyrosine kinase arrangements in activation or autophosphorylation of receptor tyrosine kinases. *Mol Cells* **29**, 443-448, doi:10.1007/s10059-010-0080-5 (2010).
- Hubbard, S. R., Mohammadi, M. & Schlessinger, J. Autoregulatory mechanisms in protein-tyrosine kinases. *J Biol Chem* **273**, 11987-11990, doi:10.1074/jbc.273.20.11987 (1998).
- Schlessinger, J. Cell signaling by receptor tyrosine kinases. *Cell* **103**, 211-225, doi:10.1016/s0092-8674(00)00114-8 (2000).
- Schlessinger, J. Receptor tyrosine kinases: legacy of the first two decades. *Cold Spring Harb Perspect Biol* **6**, doi:10.1101/cshperspect.a008912 (2014).

- Opatowsky, Y. *et al.* Structure, domain organization, and different conformational states of stem cell factor-induced intact KIT dimers. *Proc Natl Acad Sci U S A* **111**, 1772-1777, doi:10.1073/pnas.1323254111 (2014).
- Yang, Y., Xie, P., Opatowsky, Y. & Schlessinger, J. Direct contacts between extracellular membrane-proximal domains are required for VEGF receptor activation and cell signaling. *Proc Natl Acad Sci U S A* **107**, 1906-1911, doi:10.1073/pnas.0914052107 (2010).
- Chung, I. *et al.* Spatial control of EGF receptor activation by reversible dimerization on living cells. *Nature* **464**, 783-787, doi:10.1038/nature08827 (2010).
- Yuzawa, S. *et al.* Structural basis for activation of the receptor tyrosine kinase KIT by stem cell factor. *Cell* **130**, 323-334, doi:10.1016/j.cell.2007.05.055 (2007).
- Krimmer, S. G. *et al.* Cryo-EM analyses of KIT and oncogenic mutants reveal structural oncogenic plasticity and a target for therapeutic intervention. *Proc Natl Acad Sci U S A* **120**, e2300054120, doi:10.1073/pnas.2300054120 (2023).

Reviewer #3: The whole manuscript, especially the figures, needs to be tidied up, data put into supplemental figures if needed, and a more tidy and streamlined article needs to be produced. Currently it feels that the authors have emptied their lab notebooks without any discretion. It does not make for easy reading or digestion by readers in the field.

Author Response: We thank the Reviewer#3 for critical comments. Based on the suggestions, following changes are made:

1. Figure 2E and 2F: Low and high VEGFR Expressing cells are indicated.
2. Figure 3: a) Bottom panels of **Figure 3C and 3D** moved to supporting Figure **S3 D-E**.
 - b) The P values for each pair of constructs (-/+ VEGF₁₆₅) are added to **Figure 3E**.
 - c) **Figure 3F-G** is moved to the supporting **Figure S3 F-G**.
3. Figure 4: a) Bottom panel of **Figure 4E** is moved to the supporting Source
 - b) The P values for each pair of constructs (-/+ VEGF) are added to **Figure 4F**.
4. Figure 5: **Figure 5F-I** are moved to supporting **Figure S6H-K**
5. Figure 6: a) Right panel in **Figure 6B-C** are moved to the source data file and the bottom panel of **Figure 6D** has been moved to supporting **Figure S7C**.
 - b) The Left panel of **Figure 6G** is moved to **Figure S7F**.
 - c) The P values for each pair of constructs (-/+ VEGF) are added to **Figure 6E**.
6. Figure 7: Pannels in **Figure 7A-F** are removed.
7. Figure S1: Figures S1B-F are moved to the source data file. Instead, two new figures **S1B** and **S1C** are added.
8. Figure S2: **Figure S2G** and bottom panel of Figure S2 G, D, and E are moved to the source data file.
9. **Figure S3:** Main **Figure 3F-G** is moved to the supporting **Figure S3 F-G**.
10. Figure S5: **Figure S5C-D** are moved to source data file. Statistical analysis added to new **panel D**.

11. Figure S6: **Figure S6B-E and J** are removed and **Figure 5F-I** from main figure is added as supporting **Figure S6F-I**.
12. Figure S7: 1. **Figure S6C-F** are moved to the source data file. The bottom panel of **Figure 6D** has been moved to supporting **Figure S7C**. The immunoblot from **Figure 6G** is moved to **Figure S7F**.
13. Figure S8: Removed
14. Removed the section on “Cellular phosphatase balance modulates VEGFR-1 basal activation”
15. The following references are removed (previous draft): Supporting information: Reference number 9
Main text: Reference No. 80, 81, 82, 83, 84, 85, and 86
16. Added new references (Current Draft): Supporting information Reference number 20 and 21
Main text: 15, 24, 25, 27, 56, 57, 76, 81-83, 85-88, 91-95, 96, 97, 100, 101.
17. Table S4: Tyrosine Phosphorylation rate and phosphorylation half-life of VEGFR constructs after treatment with Sodium orthovanadate are removed.
18. Rewritten the Discussion section.

-----Response to Reviewer # 3 End-----

Reviewer #1 (Remarks to the Author):

The authors have addressed all of the issues raised by my initial review. Specifically:

1. Anomalies / inconsistencies in the correlation between receptor levels and tyrosine phosphorylation have been comprehensively covered.
2. Overall, the 'density' of the paper has been reduced, improving its overall focus on the key finding and readability / clarity.
3. Substantial amounts of the more peripheral data has been moved to Supplementary. I'm still not entirely convinced that there is any need for the MD simulation, even as a speculative discussion point, but it's certainly not a 'deal breaker'. It does offer some framework for discussion of a potential molecular mechanism, and it is made clear that this is speculative.

My overall view is that the paper is now substantially improved and can be published in the revised form. I think it will be of significant interest to researchers looking at RTK structure and function.

Reviewer #2 (Remarks to the Author):

The authors did a great job responding to the reviewer critique. There are a few minor updates recommended:

VEGFR3 is not the primary receptor for VEGFs, it is the primary lymphangiogenic receptor, so this update that was made to the introduction is requested to be modified to better represent the VEGFR3 role.

In the Result section, on Page 9, the authors share: "The strongly polarized electrostatic surface of the ECD (D4-D7) is likely the dominant negative regulator of VEGFR1 activation that prevents receptor dimerization in the absence of ligand 24,65,67.". They mutated C471R (which is in D5) but autophosphorylation of VEGFR1 is not clearly shown--so this conclusion is not clear. It is recommended that this instead be updated--perhaps to say that autophosphorylation of VEGFR1 is related to electrostatic polarization in the ECD, not to one cysteine.

Reviewer #3 (Remarks to the Author):

The revised manuscript has been substantially revised. In addressing the comments carefully by all 3 referees, the authors have now improved the quality of the study. This is a very good study which lends new insights into RTK signaling and biological function in multicellular eukaryote organisms.

Response to Reviewer #2:

Reviewer # 2: The authors did a great job responding to the reviewer critique. There are a few minor updates recommended:

VEGFR3 is not the primary receptor for VEGFs, it is the primary lymphangiogenic receptor, so this update that was made to the introduction is requested to be modified to better represent the VEGFR3 role.

Author Response: We thankfully acknowledge the positive comments made by the reviewer. Based on the suggestions, we now changed the first paragraph of the introduction section. All the changes are highlighted with cyan.

Reviewer # 2: In the Result section, on Page 9, the authors share: "The strongly polarized electrostatic surface of the ECD (D4-D7) is likely the dominant negative regulator of VEGFR1 activation that prevents receptor dimerization in the absence of ligand 24,65,67.". They mutated C471R (which is in D5) but autophosphorylation of VEGFR1 is not clearly shown--so this conclusion is not clear. It is recommended that this instead be updated--perhaps to say that autophosphorylation of VEGFR1 is related to electrostatic polarization in the ECD, not to one cysteine.

Author Response: We have made changes in paragraph two of page 9 as suggested by the Reviewer. All the changes are highlighted in cyan.

-----Response to Reviewer # 2 End-----